# The third pillar of causal analysis? A measurement perspective on causal representations

**Dingling Yao**[*1], **Shimeng Huang**[*1], **Riccardo Cadei**[1], **Kun Zhang**[2,3], and **Francesco Locatello**[1]

[1]Institute of Science and Technology Austria
[2]Carnegie Mellon University
[3]Mohamed bin Zayed University of Artificial Intelligence (MBZUAI)
[*]*Equal contribution.*

## Abstract

Causal reasoning and discovery, two fundamental tasks of causal analysis, often face challenges in applications due to the complexity, noisiness, and high-dimensionality of real-world data. Despite recent progress in identifying latent causal structures using causal representation learning (CRL), what makes learned representations useful for causal downstream tasks and how to evaluate them are still not well understood. In this paper, we reinterpret CRL using a measurement model framework, where the learned representations are viewed as proxy measurements of the latent causal variables. Our approach clarifies the conditions under which learned representations support downstream causal reasoning and provides a principled basis for quantitatively assessing the quality of representations using a new Test-based Measurement EXclusivity (T-MEX) score. We validate T-MEX across diverse causal inference scenarios, including numerical simulations and real-world ecological video analysis, demonstrating that the proposed framework and corresponding score effectively assess the identification of learned representations and their usefulness for causal downstream tasks. Reproducible code can be found at https://github.com/shimenghuang/a-measurement-perspective-of-crl.

## 1 Introduction

Causal analysis rests on two foundational pillars: causal reasoning and causal discovery. Causal reasoning operates under the assumption that the causal structure is known or can be assumed, and leverages data to make quantitative causal statements, for example, about the average effect of one variable on another. As causal structures are often unknown, causal discovery aims to uncover this structure, assuming that the causal variables of interest are readily observed. In many real-world settings, however, the causal variables may not be directly observable. While originally formulated mostly to enable causal capabilities in machine learning models, Causal Representation Learning (CRL, Schölkopf et al., 2021) has the potential to serve as a third pillar of causal analysis: enabling applications of causality involving unstructured data. For this, we reinterpret causal representation learning using the formalism of "*measurement models*" (Silva et al., 2006), wherein the learned representations serve as proxy measurements for latent causal variables. This perspective of CRL allows us to better characterize when a representation supports downstream causal reasoning, and it also provides a principled basis for quantitatively evaluating the quality of identification.

Methodologically, CRL tackles a more challenging task compared to independent component analysis (ICA) and disentanglement, where the latent variables are assumed to be independent of each other (Hyvärinen and Pajunen, 1999; Hyvarinen et al., 2019; Higgins et al., 2017; Locatello et al., 2019). Instead, CRL aims to unmix a set of causally related latent variables. Many recent causal representation learning works have provided different theoretical results for causal variable

39th Conference on Neural Information Processing Systems (NeurIPS 2025).

identification compiling various problem settings (von Kügelgen et al., 2021, 2024; Zhang et al., 2024b; Ahuja et al., 2024, 2022; Varici et al., 2024; Zhang et al., 2024a; Yao et al., 2024b; Kong et al., 2022; Lippe et al., 2022b; Xie et al., 2024; Dong et al., 2024; Lachapelle et al., 2022, 2023; Yao et al., 2022; Zhang et al., 2024a; Squires et al., 2023; Buchholz et al., 2024; Kong et al., 2023), recently unified by (Yao et al., 2025) into a single general methodology. Although most of the results have been theoretical in nature, machine learning models explicitly empowered with identified causal structure have been shown to be more robust under distributional shifts and provide better out-of-distribution generalization (Fumero et al., 2024; Ahuja et al., 2021; Bareinboim and Pearl, 2016; Zhang et al., 2020; Rojas-Carulla et al., 2018). From an AI for science perspective, CRL has shown its potential in understanding climate physics from raw measurement data (Yao et al., 2024a), answering causal questions in the scope of ecology experiments (Cadei et al., 2024, 2025; Yao et al., 2025), psychometric studies (Dong et al., 2024), and countless more applications related to biomedicine (Zhang et al., 2024a; Sun et al., 2025; Ravuri et al., 2025; Jain et al., 2024).

Despite recent progress in identifying latent causal structures within causal representation learning, it remains unclear what makes learned representations useful for downstream causal tasks and how to best evaluate them. Building on the proposed measurement model framework, we introduce a new evaluation metric, the Test-based Measurement EXclusivity (T-MEX) Score, which effectively quantifies how well the learned representation aligns with the underlying measurement model. This underlying measurement model can be specified by, for instance, identifiability theory of a CRL algorithm (Fig. 1), assumptions for a particular causal reasoning task (Figs. 2 and 4), or ground truth knowledge. In contrast to commonly used CRL evaluation metrics, which suffer from clear limitations (§ 4), we demonstrate that T-MEX reliably assesses both the identifiability (Defn. B.1) and causal validity (Defn. 2.2) of learned representations, as shown in a wide range of causal reasoning tasks across numerical simulations and real-world ecological video analysis (§ 5). We summarize the main contributions of this paper as follows:

- We reinterpret CRL using a *measurement model* framework, wherein the learned representations serve as proxy measurements for latent causal variables (§ 2). This formalism provides a clearer characterization of both the identification quality of learned representation and its usefulness for causal downstream tasks.

- We propose a new evaluation metric (T-MEX) that quantifies the alignment of the representations and the underlying measurement model (§ 3), and we demonstrate its advantages over widely used CRL evaluation metrics that suffer from notable limitations (§ 4).

- Supported by theoretical analysis, our empirical evaluations confirm that T-MEX maintains validity and effectiveness across diverse causal reasoning scenarios, including treatment effect estimation and covariate adjustment in both numerical simulations and real-world ecological experiments (§ 5).

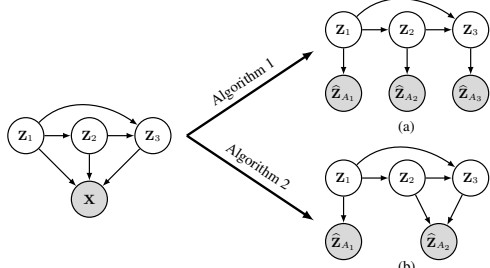

Figure 1: (*Left*) A measurement model where $\mathbf{X}$ is a fully mixed measurement of the causal variables. $\mathbf{X}$ is often termed the *observables* in CRL literature, representing the observed data. (*Right*) Two measurement models specified by different CRL identification algorithms: (a) Algorithm 1 guarantees one-to-one correspondence between the learned representation and causal variables; (b) Algorithm 2 guarantees that $\widehat{\mathbf{Z}}_{A_1}$ corresponds to $\mathbf{Z}_1$ while $\widehat{\mathbf{Z}}_{A_2}$ represents a mixing of $\mathbf{Z}_2$ and $\mathbf{Z}_3$.

## 2 CRL from A Measurement Model Perspective

**Notation.** Throughout, we write $[N]$ as shorthand for the set $\{1, \dots, N\}$. Random vectors are denoted by bold uppercase letters (e.g. $\mathbf{Z}$) and their realizations by bold lowercase (e.g., $\mathbf{z}$), indexed by superscripts. For instance, $n$ samples of $\mathbf{Z}$ are written as $\{\mathbf{z}^k\}_{k \in [N]}$. A vector $\mathbf{Z}$ can be sliced either by a single index $i \in [\dim(\mathbf{Z})]$ via $\mathbf{Z}_i$ or a index subset $A \subseteq [\dim(\mathbf{Z})]$ with $\mathbf{Z}_A := \{\mathbf{Z}_i : i \in A\}$. $P_{\mathbf{Z}}$ denotes the probability distribution of the random vector $\mathbf{Z}$ and $p_{\mathbf{Z}}(\mathbf{z})$ denotes the associated probability density function (We omit the subscription and write $p(\mathbf{z})$ when the context is clear). By default, a "measurable" function is *measurable* w.r.t. the Borel sigma algebras and is defined w.r.t. the Lebesgue measure. A more comprehensive summary of notations is provided in App. A.

## 2.1 The Measurement Model Framework

We formulate causal representation learning using a measurement model framework inspired by the formalism of (Silva et al., 2006).

**Definition 2.1** (Measurement model). Let $\mathbf{V} = (\mathbf{Z}, \widehat{\mathbf{Z}})$ be a collection of variables that can be partitioned into two sets: a set of latent *causal variables* $\mathbf{Z} = \{\mathbf{Z}_1, \cdots, \mathbf{Z}_N\}$ with $\mathbf{Z}_i$ taking values in $\mathbb{R}$ for all $i \in [N]$, and a set of observed *measurement variables* $\widehat{\mathbf{Z}} = \{\widehat{\mathbf{Z}}_{A_1}, \cdots, \widehat{\mathbf{Z}}_{A_M}\}$ where for all $j \in [M]$, $\widehat{\mathbf{Z}}_{A_j}$ takes values in $\mathbb{R}^{D_j}$ with $D_j \in \mathbb{N}_+$, and it holds that $\widehat{\mathbf{Z}} \cap \mathbf{Z} = \varnothing$.

A *measurement model* $\mathcal{M} = \langle \mathbf{Z}, \widehat{\mathbf{Z}}, \{h_j\}_{j=1}^M \rangle$ specifies that $\widehat{\mathbf{Z}}$ follows a deterministic structural causal model

$$\left\{ \widehat{\mathbf{Z}}_{A_j} \coloneqq h_j(\mathbf{Z}_{\mathrm{pa}(\widehat{\mathbf{Z}}_{A_j})}) \right\}_{j=1}^M,$$

where $\mathrm{pa}(\widehat{\mathbf{Z}}_{A_j}) \subseteq [N]$ for all $j \in [M]$, and $\mathbf{Z}_{\mathrm{pa}(\widehat{\mathbf{Z}}_{A_j})} \subseteq \mathbf{Z}$ are called the causal parents of $\widehat{\mathbf{Z}}_{A_j}$. The functions $h_j$ for all $j \in [M]$ are called the *measurement functions*. If for some $j \in [M]$, $|\mathrm{pa}(\widehat{\mathbf{Z}}_{A_j})| = 1$ and the function $h_j$ is the identity map, then the causal variable $\mathrm{pa}(\widehat{\mathbf{Z}}_{A_j})$ is said to be *measured directly*. ♣

*Remark* 2.1 (Difference from (Silva et al., 2006)). While we borrow the concept of a measurement model from Silva et al. (2006), our framework differs in two key aspects. First, Silva et al. (2006) aims to uncover relationships among latent causal variables by searching for pure measurements, i.e., a tree-structure in which latent nodes have fixed, noisy, low-dimensional observed children (measurements). In contrast, we interpret a given causal representation produced by a CRL algorithm as measurement variables and focus on evaluating their usefulness for specific causal tasks, which requires specification of a causal model. Second, Silva et al. (2006) assumes a linear latent structural causal model, whereas our framework imposes no parametric structural assumption on the latent causal variables. Rather, we specify the relationship between the causal variables and their measurements according to certain hypotheses, such as identification guarantees, prior knowledge, or assumptions for specific causal downstream tasks. As we will see in § 3, this also allows us to properly evaluate a learned CRL model. ♠

*Remark* 2.2. While we treat the measurement variables $\widehat{\mathbf{Z}}$ as noise-free nonlinear mixing of their causal parents, we can easily extend our framework to noisy measurements by considering the noise variables as additional latent causal variables. ♠

**Example 2.1.** Assume by the identifiability theory of a specific CRL method that each $\widehat{\mathbf{Z}}_{A_j}$ block-identifies (see Defn. B.1 (von Kügelgen et al., 2021, Defn 4.1)) a subset of latent variables $\mathbf{Z}_{S_i}$ ($S_i \subseteq [N]$). Then for the measurement model $\mathcal{M} = \langle \mathbf{Z}, \widehat{\mathbf{Z}}, \{h_j\}_{j=1}^M \rangle$ it holds that: $\widehat{\mathbf{Z}}_{A_j} \coloneqq h_j(\mathbf{Z}_{S_i})$, with $h_j : \mathbb{R}^{|S_i|} \to \mathbb{R}^{D_j}$ a diffeomporphism for all $j \in [M]$.

The measurement model induces a partial directed acyclic graph (DAG), that is, for any latent variable $q$ that is block-identified (Defn. B.1) by $A_j$, there is an edge from the latent causal variable $\mathbf{Z}_q$ to the measurement variable $\widehat{\mathbf{Z}}_{A_j}$, and the measurement function $h_j$ is a diffeomorphism. Illustrative examples are shown in Fig. 1 for different identifiability guarantees. ♦

**Discussion.** Note that a measurement model specified by certain identifiability theory (see Fig. 1) is a necessary but not sufficient condition for drop-in replacement of a variable with its identified counterpart in a causal inference engine (Pearl and Mackenzie, 2018) or a downstream causal estimand like *average treatment effect* (Robins et al., 1994). To this end, we introduce *causally valid measurement model*.

**Definition 2.2** (Causally valid measurement model). The measurement model (Defn. 2.1) is *"causally valid"* with respect to a statistical estimand $g$ that identifies a target causal estimand, if the measurement $\widehat{\mathbf{Z}}$ is a drop-in replacement in $g$ for the true causal variables $\mathbf{Z}$, i.e., $g(\mathbf{Z}) = g(\widehat{\mathbf{Z}})$. ♣

**Discussion.** Causal validity of a measurement model with respect to a specific estimand boils down to the estimand being invariant with respect to the measurement function. As (von Kügelgen et al., 2024) already pointed out, identification of a latent causal variable up to a non-linear parameterization (i.e., block-identifiability (Defn. B.1)) does not allow average treatment effect estimation if either the treatment or outcome is a latent causal variable without additional information. For that, a

direct measurement (see Defn. 2.1) as in (Cadei et al., 2024, 2025) is necessary; alternatively, one can choose an estimand that is invariant to non-linear invertible parameterizations, e.g., (conditional) mutual information (Janzing et al., 2013). As another example, a non-linear invertible parameterization is enough to model confounding variables (Yao et al., 2024a) and instruments, see F for extended discussions and examples. Finally, note that the causal validity of the measurement models does not always require one-to-one correspondence between the measurement variables and latent causal variables: When an estimand concerns a coarse-graining of a subset of variables, then a measurement model mixing the right subset of variables can still be causally valid. For example, the valid adjustment set $\mathbf{W}$ in Fig. 11 can contain two or more variables, which can remain entangled with each other in the learned representation $\widehat{\mathbf{W}} := h(\mathbf{W})$ as long as the measurement function $h$ is invertible, see App. F for detailed derivations.

**When is a measurement model "true"?** Note that any causal model between learned representation can always be trivially formulated as a measurement model, with each identified representation variable corresponding to a latent causal variable (i.e., $\widehat{\mathbf{Z}}_1 \to \widehat{\mathbf{Z}}_2$ implicitly implies a measurement model $\widehat{\mathbf{Z}}_1 \leftarrow \mathbf{Z}_1 \to \mathbf{Z}_2 \to \widehat{\mathbf{Z}}_2$). Sometimes, by means of other assumptions, the latent causal model may not match one-to-one with the measurements; for example, see Fig. 1 (b). Our discussion on the measurement model only specifies the dependency between a learned representation and an (implicitly) assumed latent causal model. Following (Peters et al., 2014), we intend the latent causal model to be true if it agrees with the results of randomized studies in practice. If the latent causal model is true, then a causally valid measurement model is trivially also true.

# 3 Evaluating Causal Representations using Measurement Models

This section explains how the measurement model formalism we introduced in § 2 serves as a natural tool to evaluate causal representations. A causal representation is defined as a set of measurement variables output from an encoder — a parameterized function that maps the observables $\mathbf{X}$ to the measurement variables $\widehat{\mathbf{Z}}$. Each CRL method specifies a measurement model, either through its identifiability guarantees or the particular causal task it addresses. This measurement model defines which causal variables a representation should *exclusively measure*. Given paired samples of the true causal variables $\mathbf{Z}$ and their corresponding measurement variables $\widehat{\mathbf{Z}}$ from a trained CRL model, evaluation boils down to comparing the measurement model against the observed joint distribution $P_{\mathbf{Z}, \widehat{\mathbf{Z}}}$. Before presenting our proposed evaluation metric, we introduce the following additional notation.

**Additional notation.** Let $\mathbf{Z}_1$, $\mathbf{Z}_2$, and $\mathbf{Z}_3$ be three absolutely continuous random variables taking values in $\mathbb{R}^{d_{Z_1}}$, $\mathbb{R}^{d_{Z_2}}$, and $\mathbb{R}^{d_{Z_3}}$ respectively. We say that $\mathbf{Z}_1$ and $\mathbf{Z}_2$ are *conditionally independent* given $\mathbf{Z}_3$ if $p(\mathbf{Z}_1, \mathbf{Z}_2 \mid \mathbf{Z}_3) = p(\mathbf{Z}_1 \mid \mathbf{Z}_3)p(\mathbf{Z}_2 \mid \mathbf{Z}_3)$, and it is denoted as $\mathbf{Z}_1 \perp\!\!\!\perp \mathbf{Z}_2 \mid \mathbf{Z}_3$. A statistical test $\varphi$ is a function that maps data to $\{0, 1\}$, e.g., $\varphi : \mathbb{R}^{n \times d_{Z_1}} \times \mathbb{R}^{n \times d_{Z_2}} \times \mathbb{R}^{n \times d_{Z_3}} \to \{0, 1\}$, where $n$ denotes the number of samples. The test $\varphi$ rejects a null hypothesis $\mathcal{H}_0$ if $\varphi(\mathbf{Z}_1, \mathbf{Z}_2, \mathbf{Z}_3) = 1$ and does not reject it if $\varphi(\mathbf{Z}_1, \mathbf{Z}_2, \mathbf{Z}_3) = 0$. Given a significance level $\alpha \in (0, 1)$, a test is said to be *valid* if it holds that $\sup_{P \in \mathcal{H}_0} \mathbb{P}(\varphi(\mathbf{Z}_1, \mathbf{Z}_2, \mathbf{Z}_3) = 1) \leq \alpha$, and it is said to have power $\beta \in (0, 1)$ against an alternative distribution $P \notin \mathcal{H}_0$ if $\mathbb{P}(\varphi(\mathbf{Z}_1, \mathbf{Z}_2, \mathbf{Z}_3) = 1) = \beta$.

**Exclusivity of measurements.** A measurement model describes the relationship between the causal and the measurement variables. Specifically, it tell us for each measurement variable, which causal variables it should *exclusively measure*. We formally define this concept below.

**Definition 3.1** (Exclusivity of a measurement variable). Let $\mathcal{M} = \langle \mathbf{Z}, \widehat{\mathbf{Z}}, \{h_j\}_{j \in [M]} \rangle$ be a measurement model, if a measurement variable $\widehat{\mathbf{Z}}_{A_j}, j \in [M]$ only has one causal parent $\mathbf{Z}_i$ for some $i \in [N]$, then we say $\widehat{\mathbf{Z}}_{A_j}$ *exclusively measures* $\mathbf{Z}_i$. ♣

Given samples of the causal and measurement variables $\{(\mathbf{z}^k, \hat{\mathbf{z}}^k)\}_{k \in [n]}$, we can check whether the measurement variables do satisfy the exclusivity property in the data by testing the following null hypotheses:

$$\mathcal{H}_0(i, j) : \widehat{\mathbf{Z}}_{A_j} \perp\!\!\!\perp \mathbf{Z}_i \mid \mathbf{Z}_{[N] \setminus \{i\}}, \tag{3.1}$$

for all $i \in [N]$ and $j \in [M]$. For a numerical summary of the overall exclusivity of the measurement variables, we propose the following *Test-based Measurement EXclusivity (T-MEX)* score.

**Definition 3.2** (Test-based measurement exclusivity score). Let $V \in \{0,1\}^{N \times M}$ be the adjacency matrix corresponding to the conditional independencies according to a measurement model $\mathcal{M}$, such that for all $j \in [M]$ and $i \in [N]$, $V_{ji} = 1$ if a causal variable $\mathbf{Z}_i$ is a causal parent of a measurement variable $\widehat{\mathbf{Z}}_{A_j}$ according to the measurement model, and $V_{ji} = 0$ otherwise. Let $\widehat{W} \in \{0,1\}^{N \times M}$ be the matrix constructed according to the test results of the conditional independencies in eq. (3.1) based on the samples of $(\mathbf{Z}, \widehat{\mathbf{Z}})$, such that for all $j \in [M]$ and $i \in [N]$, $W_{ji} = 1$ if $\mathcal{H}_0(i,j)$ is rejected, and $W_{ji} = 0$ otherwise. Then the test-based measurement exclusivity (T-MEX) score is defined as the *hamming distance* between $V$ and $\widehat{W}$:

$$\text{T-MEX}(V, \widehat{W}) := \sum_{j=1}^{M} \sum_{i=1}^{N} \mathbb{1}(V_{ji} \neq \widehat{W}_{ji}),$$

where $\mathbb{1}$ denotes the indicator function. ♣

Details for computing T-MEX is given in Alg. 1. As T-MEX score is based on conditional independence testing, its value depends on the randomness in the samples, and the properties of the statistical tests being used. In Prop. 3.1, we show the upper bound of the expected T-MEX score when the joint distribution $P_{\mathbf{Z}, \widehat{\mathbf{Z}}}$ of the causal variables $\mathbf{Z}$ and output measurement variables $\widehat{\mathbf{Z}}$ from a CRL model does align with a measurement model.

**Proposition 3.1.** *Let $\{\varphi_{ij}\}_{i \in [N], j \in [M]}$ be a family of tests for eq. (3.1) where for all $i \in [N]$ and $j \in [M]$, $\varphi_{ij}$ is valid with level $\alpha \in (0,1)$ and has power at least $\beta \in (0,1)$. Given an adjacency matrix $V \in \mathbb{R}^{N \times M}$ based on a measurement model, if the joint distribution $P_{\mathbf{Z}, \widehat{\mathbf{Z}}}$ of the causal and measurement variables does align with the measurement model, and each entry in $\widehat{W}$ is computed based on an independent set of samples $\{(\mathbf{z}^k, \hat{\mathbf{z}}^k)\}_{k \in [n_{ij}]}, n_{ij} \in \mathbb{N}_+$, then the expected T-MEX satisfies*

$$\mathbb{E}[\text{T-MEX}(V, \widehat{W})] \leq \alpha \cdot (MN - ||V||_1) + (1 - \beta) \cdot ||V||_1,$$

*where $||V||_1 = \sum_{i=1}^{N} \sum_{j=1}^{M} V_{ij}$ is the $L_1$-norm of $V$.*

*Remark* 3.1. Prop. 3.1 assumes that each null hypothesis in eq. (3.1) is tested using an independent set of samples. When there is only one set of samples available for a large number of tests, using the same sample set can lead to inflation of the false positive rate, and may inflate the T-MEX score. In this case, we recommend doing a multiple comparison adjustment when constructing $\widehat{W}$, for example, the Bonferroni-Holm correction (Holm, 1979), which controls the family-wise error rate while it does not make assumptions on the dependencies of the multiple p-values. ♠

*Remark* 3.2. In this section, we focus on the exclusivity perspective of a measurement model via an approach similar to the idea of falsification of causal graphs (e.g., Kook, 2025; Faller et al., 2024). This is a non-parametric approach which is agnostic to the measurement functions. In certain cases, however, a measurement model may contain not only the conditional independence structure, but also other parametric assumptions through specifications of the measurement functions $\{h_j\}_{j \in [M]}$. Then, one may extend T-MEX to also take these constraints into account. ♠

## 4 Related Work: Flaws of Existing Evaluation Metrics for CRL

In this section, we cover the metrics that have been used by most papers proposing causal representation learning approaches (von Kügelgen et al., 2021, 2024; Zheng et al., 2022; Ahuja et al., 2024, 2022; Varici et al., 2024; Zhang et al., 2024a,b; Yao et al., 2024b; Lippe et al., 2022a,b; Lachapelle et al., 2022, 2023; Yao et al., 2022; Zhang et al., 2024a; Squires et al., 2023; Buchholz et al., 2024; Yao et al., 2025) to name a few. We highlight how these metrics are not immediately suitable to evaluate identification results in the presence of causal relations, making it difficult to compare models and requiring great care in the interpretation of the results that is often missed (Gamella et al., 2025).

Standard evaluation for latent variable identification in existing CRL works employs *coefficient of determination $R^2$* (Defn. 4.1), and *mean correlation coefficient* (Defn. 4.2). However, when the latent variables are causally related, a high score of these two metrics does not indicate that the learned representations align with the measurement model we expect from the identifiability theory. Example 4.1 illustrates this limitation of these two metrics under the presence of causal dependencies.

**Example 4.1.** Assume that the latent causal variables $\mathbf{Z}$ in Fig. 1 (b) follow a linear Gaussian additive noise model. Specifically, the latent variables $\mathbf{Z}_1$ and $\mathbf{Z}_2$ are generated based on the following structural equation:

$$\mathbf{Z}_2 := a \cdot \mathbf{Z}_1 + e \tag{4.1}$$

with $e \sim P_e$, $\mathbb{E}[e] = 0$ and $e \perp\!\!\!\perp \mathbf{Z}_1$. Suppose that the measurement model which induces Fig. 1 (b) specifies that the measurement function $h : \mathbb{R} \rightarrow \mathbb{R}$ is a diffeomorphism such that $\widehat{\mathbf{Z}}_{A_1} = h(\mathbf{Z}_1)$, that is, $\widehat{\mathbf{Z}}_{A_1}$ identifies $\mathbf{Z}_1$, while $\widehat{\mathbf{Z}}_{A_1}$ should not contain any additional information about $\mathbf{Z}_2$. ◆

**Coefficient of determination.** $R^2$ measures the proportion of the variation in the dependent variables explained by the regression model (Draper and Smith, 1998), formally defined as

**Definition 4.1** (Population $R^2$ score). Let $(\mathbf{Z}_i, \widehat{\mathbf{Z}}_{A_j})$ be a pair of random variables both taking values in $\mathbb{R}$, $i \in [N], j \in [M]$. The coefficient of determination $R^2$ score for predicting $\mathbf{Z}_i$ from $\widehat{\mathbf{Z}}_{A_j}$ is defined as

$$R^2(\mathbf{Z}_i, \widehat{\mathbf{Z}}_{A_j}) := \frac{\mathbb{V}(\mathbb{E}[\mathbf{Z}_i \mid \widehat{\mathbf{Z}}_{A_j}])}{\mathbb{V}(\mathbf{Z}_i)},$$

where $\mathbb{E}$ and $\mathbb{V}$ denote the expectation and variance operators, respectively. ♣

**Problem of $R^2$ in Example 4.1**: Let $R^2(\mathbf{Z}_1, \widehat{\mathbf{Z}}_{A_1})$ denote the $R^2$ score as defined in Defn. 4.1. Following the linear mechanism in eq. (4.1), $R^2(\mathbf{Z}_2, \widehat{\mathbf{Z}}_{A_1})$ can be expressed as

$$
\begin{aligned}
R^2(\mathbf{Z}_2, \widehat{\mathbf{Z}}_{A_1}) &= \frac{\mathbb{V}(\mathbb{E}[\mathbf{Z}_2 \mid \widehat{\mathbf{Z}}_{A_1}])}{\mathbb{V}(\mathbf{Z}_2)} = \frac{\mathbb{V}(\mathbb{E}[a\mathbf{Z}_1 + e \mid \widehat{\mathbf{Z}}_{A_1}])}{\mathbb{V}(a\mathbf{Z}_1 + e)} \\
&= \frac{a^2 \mathbb{V}(\mathbb{E}[\mathbf{Z}_1 \mid \widehat{\mathbf{Z}}_{A_1}])}{a^2 \mathbb{V}(\mathbf{Z}_1) + \mathbb{V}(e)} = \frac{a^2 \mathbb{V}(\mathbf{Z}_1)}{a^2 \mathbb{V}(\mathbf{Z}_1) + \mathbb{V}(e)} R^2(\mathbf{Z}_1, \widehat{\mathbf{Z}}_{A_1}).
\end{aligned}
\tag{4.2}
$$

Depending on the noise level $\mathbb{V}(e)$, $R^2(\mathbf{Z}_2, \widehat{\mathbf{Z}}_{A_1})$ can be either close to $R^2(\mathbf{Z}_1, \widehat{\mathbf{Z}}_{A_1})$ when $\mathbb{V}(e) \ll a^2 \mathbb{V}(\mathbf{Z}_1)$ or close to 0 when $\mathbb{V}(e)$ is significantly higher than $a^2 \mathbb{V}(\mathbf{Z}_1)$; in either case it does not reflect whether $\widehat{\mathbf{Z}}_{A_1}$ identifies $\mathbf{Z}_2$ or not, in the sense of Defn. B.1. Ultimately, $R^2$ is a metric for predictability, not for identifiability. Using it as an identifiability metric under causal dependency can lead to misinterpretation (Gamella et al., 2025).

*Remark* 4.1 (Other problems of $R^2$ score). $R^2$ is designed to measure how well a *linear* model fits between two random variables. When the fitted model is nonlinear, $R^2$ can yield values outside $[0, 1]$, which can be misleading. See also Cameron and Windmeijer (1997) for more details. ♠

**Mean correlation coefficient (MCC).** Intuitively, MCC measures the *component-wise correspondence* between the learned representation $\widehat{\mathbf{Z}}$ and the ground truth latent variables $\mathbf{Z}$. When using MCC, it is required to have the same latent and encoding dimensions. We restate the definition of the MCC as follows.

**Definition 4.2** (Mean correlation coefficient).

$$\text{MCC} = \frac{1}{N} \max_{\pi \in \text{perm}[N]} \sum_{i=1}^{N} |\text{Corr}(\mathbf{Z}_i, \widehat{\mathbf{Z}}_{\pi(i)})|,$$

where $\text{Corr}(\cdot, \cdot)$ refers to the Pearson correlation under linear relationship and Spearman correlation in the nonlinear case. ♣

However, we notice that MCC cannot capture how well the representations are *disentangled*, misaligning with its original purpose of measuring *component-wise correspondence*. Assume in Fig. 1 (b) that $\widehat{\mathbf{Z}}_{A_1} = \widehat{\mathbf{Z}}_1$ and $\widehat{\mathbf{Z}}_{A_2} = [\widehat{\mathbf{Z}}_2, \widehat{\mathbf{Z}}_3]$. The learned representations $\widehat{\mathbf{Z}}_{A_j}$ are linear mappings of their causal parents $\mathbf{Z}_{\text{pa}(\widehat{\mathbf{z}}_{A_j})}$:

$$\widehat{\mathbf{Z}}_1 = s \cdot \mathbf{Z}_1; \qquad \widehat{\mathbf{Z}}_2 = a \cdot \mathbf{Z}_2 + b \cdot \mathbf{Z}_3; \qquad \widehat{\mathbf{Z}}_3 = c \cdot \mathbf{Z}_2 + d \cdot \mathbf{Z}_3,$$

where $s, a, b, c, d \neq 0$. In this case, the MCC would obtain the highest value 1, although $\mathbf{Z}_2, \mathbf{Z}_3$ are still entangled in the learned representation $\widehat{\mathbf{Z}}$, demonstrating that MCC is inadequate in evaluating element-wise identification under causal relations.

**Evaluation of causal relations.** Causal relations are usually evaluated with the standard metrics *Structural Hamming distance* (SHD). We remark that evaluating causal discovery on the learned representations should always be done in conjunction with latent variable identification, as it is possible to achieve a perfect SHD (i.e, zero) with entangled representations, using e.g., LiNGAM (Shimizu et al., 2006), as shown numerically in App. D.3.

**Evaluation of disentangled representation.** Evaluating disentangled representations (where the ground truth latent variables are assumed to be mutually independent) is comparatively easier. In the disentangled case, the main objective is to assess how well the learned representation aligns one-to-one with the ground truth latents. Commonly used evaluation metrics for disentangled representations include the BetaVAE Score (Higgins et al., 2017), FactorVAE Score (Kim and Mnih, 2018), Mutual Information Gap (MIG Chen et al. (2018)), DCI-disentanglement (Eastwood and Williams, 2018), Modularity (Ridgeway and Mozer, 2018) and SAP (Kumar et al., 2017). Broadly, evaluating learned representations can be viewed as a two-stage procedure, first estimating the relationship between latent variables and representations, and then aggregating this information into a single score (Locatello et al., 2020). In some way, our test can be seen as following the same strategy, although evaluating variable-level correspondence is less straightforward given underlying causal relationships, making it a fundamentally more challenging and understudied problem.

## 5 Experiments

This section demonstrates the validity of the proposed T-MEX score in various causal reasoning settings. We first focus on *covariate adjustment* in numerical simulations, using T-MEX to evaluate both identifiability (Defn. B.1) and causal validity (Defn. 2.2) of the representations (§ 5.1). Next, we move on to *treatment effect estimation* in high-dimensional ecological video analysis, where we demonstrate that T-MEX effectively characterizes how well the learned representation supports answering downstream causal questions (§ 5.2). For both experiments, we estimate T-MEX based on the projected covariance measure (PCM) test (Lundborg et al., 2024) implemented in the python package `pycomets` (Huang and Kook, 2025), which is an algorithm-agnostic test for conditional independence (see App. E for more explanations). Further experiment details and additional results are deferred to App. D.

### 5.1 Numerical Simulation

This experiment validates our proposed T-MEX evaluation metric through a controlled numerical simulation. We leverage CRL to model confounders and perform backdoor adjustment to estimate the average treatment effect (ATE). We report both $R^2$ and the ATE bias, demonstrating that T-MEX closely aligns with the absolute ATE bias and provides a reliable measure of representation quality, whereas $R^2$ fails to yield consistent or meaningful conclusions.

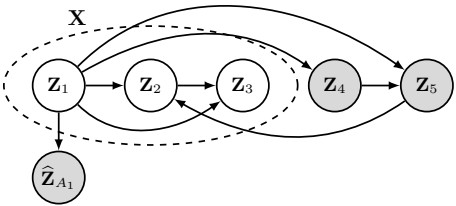

Figure 2: Measurement model containing the *latent* causal variables $\mathbf{Z}_1$, $\mathbf{Z}_2$, and $\mathbf{Z}_3$ (white nodes) and *observed* (also termed "*directly measured*" in Defn. 2.1) causal variables $\mathbf{Z}_4$ and $\mathbf{Z}_5$ (gray nodes). The entangled observable $\mathbf{X}$ is shown as a dashed oval. $\widehat{\mathbf{Z}}_{A_1}$ denotes the exclusive measurement (Defn. 3.1) of $\mathbf{Z}_1$.

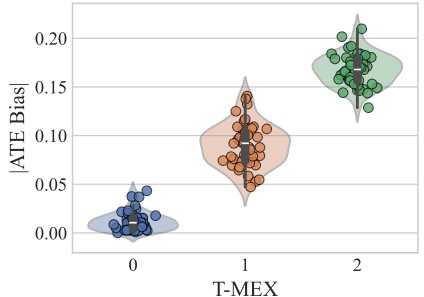

Figure 3: *T-MEX tracks the absolute bias of the ATE estimates* of $\mathbf{Z}_4$ on $\mathbf{Z}_5$ where $\widehat{\mathbf{Z}}_1$ is conditioned on as the back door adjustment.

**Experiment settings.** We generate five causal variables, $\mathbf{Z}_i$ for $i \in [5]$ according to a linear structural causal model (see App. D.1), where two of the causal variables, $\mathbf{Z}_4$ and $\mathbf{Z}_5$, are *observed* (also termed "*directly measured*" in Defn. 2.1). The entangled observations $\mathbf{X} := f(\mathbf{Z}_1, \mathbf{Z}_2, \mathbf{Z}_3)$ are

generated by applying a diffeomorphism $f : \mathbb{R}^3 \to \mathbb{R}^3$, implemented as an invertible MLP, on the causal variables. Our *target causal task is to estimate the ATE of $\mathbf{Z}_4$ on $\mathbf{Z}_5$*. As the true causal relationship between $\mathbf{Z}_4$ and $\mathbf{Z}_5$ is linear, we can construct a consistent causal estimator where $\mathbf{Z}_1$ is adjusted using linear regression, which is invariant up to *bijective transformations* of $\mathbf{Z}_1$ (App. F). Although $\mathbf{Z}_1$ is latent and cannot be directly adjusted for, one can *measure* it through a bijective transformation $\widehat{\mathbf{Z}}_{A_1} := h(\mathbf{Z}_1)$ which is obtained from the entangled observation $\mathbf{X}$. Note that in this case, $\widehat{\mathbf{Z}}_{A_1}$ *exclusively measures* (Defn. 3.1) the confounder $\mathbf{Z}_1$, as depicted in Fig. 2. We train three different CRL models based on the identifiable learning algorithm proposed by Yao et al. (2024b) and obtain samples of the measurement variable $\widehat{\mathbf{Z}}_{A_1}$:

- **Model A**: a sufficiently trained model from which we expect the learned representation $\widehat{\mathbf{Z}}_{A_1}^A$ (where by a slight abuse of notation, the superscript represents the model indicator) to *exclusively measure* $\mathbf{Z}_1$;

- **Model B**: an insufficiently trained model with unclear latent-measurement correspondence;

- **Model C**: a corrupted version of Model A where the representation $\widehat{\mathbf{Z}}_{A_1}^C$ is defined as a linear mixing of the identified representation $\widehat{\mathbf{Z}}_{A_1}^A$ and $\mathbf{Z}_2, \mathbf{Z}_3$.

**Results.** Tab. 1 summarizes the T-MEX scores together with the coefficient of determination $R^2$ for all three models A, B and C, presented as `mean±sd`. *For statistical validity*, we compute the results using 50 simulated datasets from each model, with each dataset containing 4096 observations. Further details about the test results are provided in App. D.1. Tab. 1 shows that a sufficiently trained model (Model A) achieves a low T-MEX score, indicating that the learned representation $\widehat{\mathbf{Z}}_{A_1}$ exclusively measures the latent variable $\mathbf{Z}_1$. In contrast, the insufficiently trained and corrupted models (Models B and C) exhibit high T-MEX scores, demonstrating misalignment between the learned representation and the hypothesized measurement model (Fig. 2). Fig. 3 presents the absolute ATE bias estimated from the learned representations of all three models. We observe a strong correlation between T-MEX and the absolute bias of the ATE, validating T-MEX as a reliable indicator of the causal validity of the learned representation (Defn. 2.2). In contrast, as shown in Tab. 1, $R^2$ is relatively high for all three latent variables, failing to show a clear correspondence with the ATE bias.

Table 1: T-MEX, $R^2$ scores, and Spearman correlation coefficients of the learned representations (presented as `mean±std`) of **model A** (sufficiently trained, i.e., $\widehat{\mathbf{Z}}_1$ exclusively measures $\mathbf{Z}_1$), **model B** (insufficiently trained model with unclear latent-measurement correspondence) and **model C** (manually corrupted representation by linearly mixing $Z_2, Z_3$ with the representation of model A) based on 50 simulated datasets, where each dataset contains 4096 observations.

| Model | T-MEX ($\downarrow$) | $R^2$ | | | Spearman Cor. Coef. | | |
|---|---|---|---|---|---|---|---|
| | | $\mathbf{Z}_1$ | $\mathbf{Z}_2$ | $\mathbf{Z}_3$ | $\mathbf{Z}_1$ | $\mathbf{Z}_2$ | $\mathbf{Z}_3$ |
| A | $0.1200 \pm 0.3283$ | $0.9984 \pm 0.0001$ | $0.7516 \pm 0.0064$ | $0.8001 \pm 0.0006$ | $1.0000 \pm 0.0000$ | $0.8568 \pm 0.0044$ | $0.8864 \pm 0.0040$ |
| B | $1.1800 \pm 0.3881$ | $0.6665 \pm 0.0078$ | $0.8305 \pm 0.0032$ | $0.8707 \pm 0.0027$ | $0.8434 \pm 0.0061$ | $0.9602 \pm 0.0017$ | $0.9908 \pm 0.0004$ |
| C | $2.0000 \pm 0.0000$ | $0.9394 \pm 0.0016$ | $0.5421 \pm 0.0096$ | $0.6627 \pm 0.0084$ | $0.9673 \pm 0.0013$ | $0.7215 \pm 0.0076$ | $0.8016 \pm 0.0062$ |

## 5.2 Real-world Ecological Experiment: ISTAnt

This experiment validates the T-MEX score on ISTAnt (Cadei et al., 2024), a real-world ecological benchmark designed for treatment effect estimation. We show a strong correlation between T-MEX and the absolute bias of the ATE, demonstrating that T-MEX can reliably evaluate the causal validity of learned representations under the challenge of high-dimensional real-world data.

**Experiment settings.** ISTAnt consists of video recordings of ant triplets with occasional grooming behavior. *The goal is to extract a per-frame representation for supervised behavior classification (grooming or not) to estimate the ATE of an intervention (exposure to a certain pathogen).*

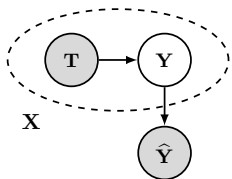

Figure 4: Measurement Model for the causal task in ISTAnt. $\mathbf{T}$ denotes the treatment (chemical exposure) and the *latent* outcome $\mathbf{Y}$ represents the ant's grooming behavior. Observable $\mathbf{X}$ (video recordings) is represented using a dashed oval. The measurement $\widehat{\mathbf{Y}}$ *exclusively measures* (Defn. 3.1) $\mathbf{Y}$.

Retrieving causally valid representations in this case is challenging as we have more non-annotated than annotated data, as described by (Cadei et al., 2024). Fig. 4 depicts the hypothesized measurement model for this particular causal task, note that the treatment $\mathbf{T}$ and outcome $\mathbf{Y}$ are unconfounded because the data is collected through a randomized controlled trial (RCT), meaning that the binary treatment $\mathbf{T}$ is randomly assigned.

**Results.** We compute the T-MEX score for 2,400 different models at a significance level of $\alpha = 0.05$, and compare both classification accuracy and absolute ATE bias against T-MEX. A full description of the considered models and training details is reported in App. D.2. We only focus on the models that yield an accuracy over 80% for meaningful statements. We observe that models with T-MEX $= 0$ achieve higher mean and lower variance for both accuracy and absolute ATE bias, demonstrating that T-MEX effectively and reliably evaluates the quality of learned representations in terms of both classification performance and causal validity (Defn. 2.2).

**Statistical validation.** To further assess the statistical significance between the T-MEX $= 0$ and T-MEX $= 1$ groups, we conduct a Mann-Whitney U test (Mann and Whitney, 1947) with the null hypothesis

$$\mathcal{H}_0 : \mathbb{E}\Big[|\text{ATE Bias}| \ \Big| \ \text{T-MEX} = 1\Big] \leq \mathbb{E}\Big[|\text{ATE Bias}| \ \Big| \ \text{T-MEX} = 0\Big].$$

The resulting p-value of $0.0047$ leads us to reject $\mathcal{H}_0$, providing strong evidence that the average absolute bias of the ATE for models with T-MEX $= 1$ is significantly higher than for those with T-MEX $= 0$. Overall, T-MEX shows a strong correlation with the absolute bias of the ATE, validating its reliability as an evaluation metric for the causal validity of learned representations (Defn. 2.2).

**Real-world implications of T-MEX.** We emphasize that the proposed T-MEX score can be computed using only observational data, possibly with selection bias, as long as this selection bias does not change the conditional independence between measurements and causal variables. Instead, calculating the ATE bias as in (Cadei et al., 2024) requires a validation set that closely approximates the underlying population of the randomized controlled trial, a significantly stronger assumption that is often difficult to satisfy in real-world settings. Overall, T-MEX offers a convenient and accessible evaluation metric that reliably quantifies the usefulness of the learned representation for a causal downstream task, without the need for additional identifying assumptions.

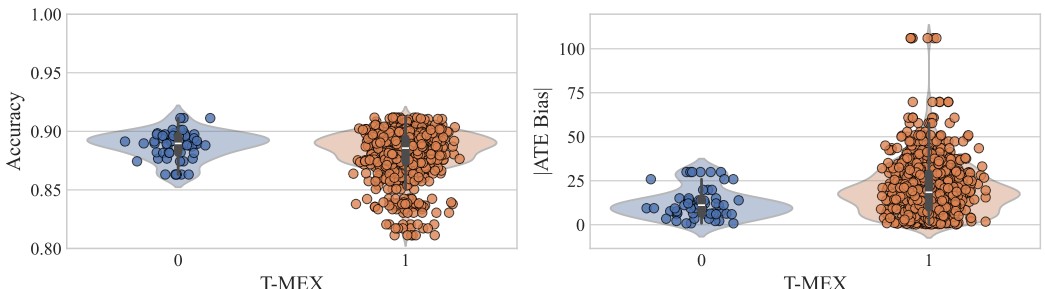

Figure 5: *T-MEX reflects model performance in terms of both classification accuracy and causal validity (Defn. 2.2).* Compared to their counterparts, models with lower T-MEX achieve consistently high accuracy (*Left*) and low absolute ATE bias with reduced variance (*Right*).

## 5.3 Evaluation of T-MEX Properties using Synthetic Data

In this section, we outline additional experiments designed to assess specific properties of T-MEX, including its scalability, robustness across different test choices, and behavior under weak or nonlinear causal relations as well as noisy measurements. The corresponding experimental details and results are provided in App. D.4.

**Reliability and scalability with higher-dimensional latent variables.** It is well known that the statistical power of conditional independence (CI) tests deteriorates as the dimensionality of the conditioning set increases—a limitation shared by most CI methods (e.g., Shah and Peters, 2020; Strobl et al., 2019; Zhang et al., 2012). We examine the reliability and scalability of T-MEX given data generated from a *non-linear location-scale SCM* with up to 50 latent nodes in Tab. 3 in App. D.4.

Since our proposed framework is agnostic to CI test choices, T-MEX can be easily scaled to larger dimensions by adapting more powerful and scalable test methods as they are developed.

**Consistency of T-MEX under different test choices**. CI testing is an important component in our framework, and choosing a valid and powerful test is essential. We chose PCM test in our experiments for its theoretical guarantees on the validity (type I error control) and power of the test, making it a reliable choice of test for the exclusivity claim. We also compare T-MEX scores using different conditional independence tests in Tab. 4 in App. D.4, based on the same experimental setup as in § 5.1. In this experiment, we see that T-MEX remains consistent across different test choices.

**Consistency of T-MEX under noisy measurements.** As discussed in Remark 2.2, noise in the measurement functions can be treated as additional independent latent variables. This means that if the noise is independent of the existing causal variables, the conditional independencies required for T-MEX remain intact. In such cases, one can safely compute T-MEX without explicitly modeling the noise; the score will remain valid. The empirical T-MEX value is shown to be robust under noise, as shown in Tab. 7 in App. D.4.

**T-MEX under weak causal relations.** Building on the experiment in § 5.1, we also examine how T-MEX behaves when the causal relationships between the latent causal variables are weak and compare it with the commonly used CRL identifiability metrics $R^2$ and MCC. See Tab. 5 in App. D.4. In this case, T-MEX correctly reflects the entanglement in the representation, while both $R^2$ and MCC give a misleading score suggesting perfect element-wise identification.

**Alignment between T-MEX and ATE under nonlinear SCM**. Extending the results in § 5.1, we further examine the correspondence between T-MEX and ATE under a nonlinear setting. As shown in Tab. 6 in App. D.4, the results are highly similar to the linear case § 5.1, T-MEX closely aligns with the absolute ATE bias, effectively evaluating causal representations for downstream inference tasks with nonlinear causal relations.

# 6 Conclusion and Limitations

This paper introduces a novel perspective on Causal Representation Learning (CRL) based on a measurement model framework, in which causal representations are treated as proxy measurements of latent causal variables (§ 2). This perspective provides a flexible framework that unites CRL identification theory with downstream task assumptions via measurement functions, yielding a principled way to evaluate representation quality. More specifically, we propose a new evaluation metric, Test-based Measurement EXclusivity (T-MEX) score, which quantifies the discrepancy between a given measurement model (specified by a CRL algorithm, a causal task, or ground truth knowledge) and the joint distribution of causal and measurement variables (representation outputs of a CRL model) using conditional independence tests (§ 3). Because these conditional independence tests impose no parametric assumptions, our T-MEX score remains broadly applicable. However, like any statistical procedure, these tests are subject to sampling variability and potential statistical errors, so the reliability of T-MEX depends on which test is chosen. By remaining agnostic about the specific test, we empower practitioners to tailor the score to whatever assumptions they are willing to make (e.g., parametric or non-parametric). We demonstrate, using both simulations (§ 5.1) and real-world video analysis (§ 5.2), that our proposed T-MEX score effectively quantifies the identification and causal validity of the learned representation (Defn. 2.2). This provides a convenient and practical evaluation scheme for representation quality in real-world scenarios, especially when the true treatment effect bias is unavailable, such as in the absence of randomized studies.

**Acknowledgment**

This research was funded in whole or in part by the Austrian Science Fund (FWF) 10.55776/COE12. For open access purposes, the author has applied a CC BY public copyright license to any accepted manuscript version arising from this submission.

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

# Appendix

## Table of Contents

## A  Notation and Terminology

This section summarizes the symbols used throughout the paper.

| | |
|---|---|
| $\mathbf{Z}$ | Causal variables |
| $\mathbf{X}$ | Observables |
| $D_j$ | Dimension of the representation $\widehat{\mathbf{Z}}_{A_j}$ |
| $N$ | Dimension of the causal variables $\mathbf{Z}$ |
| $n$ | Number of samples for the statistical tests for T-MEX |
| $\mathbb{P}(\cdot)$ | Probability operator |
| $\mathbb{E}(\cdot)$ | Expectation operator |
| $\mathbb{1}(\cdot)$ | Indicator function |

## B  Preliminaries

**Definition B.1** (Block-identifiability (von Kügelgen et al., 2021))**.** A set of latent variables $\mathbf{Z} \in \mathbb{R}^{d_z}$ is block-identified by a representation $\widehat{\mathbf{Z}} \in \mathbb{R}^{d_{\hat{z}}}$ if there exists a bijection $h : \mathbb{R}^{d_z} \rightarrow \mathbb{R}^{d_{\hat{z}}}$ such that $\widehat{\mathbf{Z}} = h(\mathbf{Z})$. ♣

## C  Proofs and Algorithms

This section includes the proof for Prop. 3.1 and the algorithm to compute the T-MEX score.

**Proposition 3.1.** *Let $\{\varphi_{ij}\}_{i \in [N], j \in [M]}$ be a family of tests for eq. (3.1) where for all $i \in [N]$ and $j \in [M]$, $\varphi_{ij}$ is valid with level $\alpha \in (0, 1)$ and has power at least $\beta \in (0, 1)$. Given an adjacency matrix $V \in \mathbb{R}^{N \times M}$ based on a measurement model, if the joint distribution $P_{\mathbf{Z}, \widehat{\mathbf{Z}}}$ of the causal and measurement variables does align with the measurement model, and each entry in $\widehat{W}$ is computed based*

**Algorithm 1:** Compute T-MEX score from one set of samples

---

**Input:** Paired samples of causal variables and measurement variables $\{\mathbf{z}, \widehat{\mathbf{z}}_{A_1}, \ldots, \widehat{\mathbf{z}}_{A_M}\}$ where
$\mathbf{z} \in \mathbb{R}^{n \times N}$ and $\widehat{\mathbf{z}}_{A_j} \in \mathbb{R}^{n \times D_j}$ for $j \in [M]$, adjacency matrix of the measurement model
$V \in \{0,1\}^{N \times M}$, a set of statistical tests for $\{\varphi_{ij}\}_{i \in [N], j \in [M]}$ for (3.1), where for all
$i \in [N], j \in [M], \varphi_{ij} : \mathbb{R}^{n \times 1} \times \mathbb{R}^{n \times D_j} \times \mathbb{R}^{n \times (N-1)} \to \{0,1\}$

**Output:** T-MEX score of the given sample

$\widehat{W} \leftarrow \mathbf{0}^{N \times M}$

**for** $i \in [N]$ **do**

    **for** $j \in [M]$ **do**

        $\widehat{W}_{ij} \leftarrow \varphi_{ij}(\mathbf{z}_i, \widehat{\mathbf{z}}_{A_j}, \mathbf{z}_{[N]\setminus\{i\}})$

    **end**

**end**

**return** $\sum_{i=1}^{N} \sum_{j=1}^{M} \mathbb{1}(V_{ij} \neq \widehat{W}_{ij})$

---

on an independent set of samples $\{(\mathbf{z}^k, \hat{\mathbf{z}}^k)\}_{k \in [n_{ij}]}, n_{ij} \in \mathbb{N}_+$, then the expected T-MEX satisfies

$$\mathbb{E}[\textit{T-MEX}(V, \widehat{W})] \leq \alpha \cdot (MN - ||V||_1) + (1 - \beta) \cdot ||V||_1,$$

where $||V||_1 = \sum_{i=1}^{N} \sum_{j=1}^{M} V_{ij}$ is the $L_1$-norm of $V$.

*Proof.* Suppose the joint distribution of $(\mathbf{Z}, \widehat{\mathbf{Z}})$ aligns with the conditional independencies indicated by the adjacency matrix $V$, that is, for all $i \in [N]$ and $j \in [M]$, if $V_{ij} = 0$, it holds that $\widehat{\mathbf{Z}}_{A_j} \perp\!\!\!\perp \mathbf{Z}_i \,|\, \mathbf{Z}_{[N]\setminus\{i\}}$; if $V_{ij} = 1$, it holds that $\widehat{\mathbf{Z}}_{A_j} \not\perp\!\!\!\perp \mathbf{Z}_i \,|\, \mathbf{Z}_{[N]\setminus\{i\}}$.

Fix a significance level $\alpha \in (0,1)$. Suppose for all $i \in [N]$ and all $j \in [M]$, the statistical test $\varphi_{ij} : \mathbb{R}^{n \times 1} \times \mathbb{R}^{n \times D_j} \times \mathbb{R}^{n \times (N-1)} \to \{0,1\}$ is valid at level $\alpha$ and has power at least $\beta \in [0,1]$ against the alternative distribution where $\widehat{\mathbf{Z}}_{A_j} \not\perp\!\!\!\perp \mathbf{Z}_i \,|\, \mathbf{Z}_{[N]\setminus\{i\}}$.

Then given independent sets of samples $\{\mathbf{z}^k, \hat{\mathbf{z}}^k\}_{k \in [n_{ij}]}$ for $i \in [N]$ and $j \in [M]$, and $\widehat{W}_{ij} = \varphi_{ij}(\mathbf{z}_i, \hat{\mathbf{z}}_{A_j}, \mathbf{z}_{[N]\setminus\{i\}})$, it holds that

- if $V_{ij} = 0$, then $P(\widehat{W}_{ij} = 1) \leq \alpha$;

- if $V_{ij} = 1$, then $P(\widehat{W}_{ij} = 0) \leq 1 - \beta$.

Therefore, the expected value of T-MEX score is given by

$$\mathbb{E}[T\text{-}MEX(V, \widehat{W})] = \alpha \cdot \sum_{i,j} \mathbb{1}(V_{ij} = 0) + (1 - \beta) \sum_{i,j} \cdot \mathbb{1}(V_{ij} = 1)$$
$$\leq \alpha \cdot (MN - ||V||_1) + (1 - \beta) \cdot ||V||_1$$

where $||V||_1$ is the 1-norm of $V$. The second inequality is implied by the that each test $\varphi_{ij}$ is valid with level $\alpha$ and has power $\geq \beta$. $\qquad\square$

*Remark* C.1. Proposition 3.1 tells us that if the measurement model does hold for the joint distribution of the causal variables and the output representations from a trained CRL model, we would expect to see a "low" T-MEX score given that we employ valid statistical tests that are also powerful enough to reject the null under alternatives. A "low" T-MEX score does not in general refer to a 0 score, as it depends on $V$, the chosen significance level $\alpha$, and the power of the test $\beta$. For example, let $\alpha = 0.05$, we consider a valid statistical test that has the highest power, i.e., $\beta = 1$, additionally, assume the number of 0s in $V$ is 2, then the expected value of the T-MEX score is no larger than $0.05 \times 2 = 0.1$. ♠

# D Experiment Details and Additional Results

This section elaborates on the experiment settings of § 5. We include further information regarding the data-generating process for the simulated experiment (§ 5.1) and the ISTAnt dataset (Cadei et al., 2024) used in the ecological case study (§ 5.2), as well as additional experimental results.

## D.1 Numerical Simulation

**Experiment setting.** We consider five causal variables $(\mathbf{Z}_1, \cdots, \mathbf{Z}_5)$ generated based on a linear structural causal model (Peters et al., 2017)

$$\mathbf{Z} = B\mathbf{Z} + \varepsilon,$$

where $\mathbf{Z} \coloneqq (\mathbf{Z}_1, \mathbf{Z}_2, \mathbf{Z}_3, \mathbf{Z}_4, \mathbf{Z}_5)$, $\mathbf{Z}$ takes values in $\mathbb{R}^5$, $\varepsilon \sim \mathcal{N}_5(0, I)$, and $B = \begin{bmatrix} 0 & 0 & 0 & 0 & 0 \\ 1 & 0 & 0 & 0 & 1 \\ 1 & 1 & 0 & 0 & 0 \\ 1 & 0 & 0 & 0 & 0 \\ 1 & 0 & 0 & 1 & 0 \end{bmatrix}$,

which induces the partial DAG depicted in Fig. 2. Two of the causal variables ($\mathbf{Z}_4$ and $\mathbf{Z}_5$) are observed (i.e., directly measured as in Defn. 2.1), and the other three ($\mathbf{Z}_1$, $\mathbf{Z}_2$, and $\mathbf{Z}_3$) are latent and we observe only a bijective mixing $\mathbf{X}$ of them.

For the purpose of latent variable identification, we consider the multiview scenario in (Yao et al., 2024b) where two views $\mathbf{X}_1, \mathbf{X}_2$ are generated from different subsets of latent variables. Formally, we have

$$\begin{aligned} \mathbf{X}_1 &= f_1(\mathbf{Z}_1, \mathbf{Z}_2) \\ \mathbf{X}_2 &= f_2(\mathbf{Z}_1, \mathbf{Z}_3), \end{aligned} \tag{D.1}$$

where $f_1, f_2 : \mathbb{R}^2 \to \mathbb{R}^2$ are diffeomorphisms, implemented using invertible MLPs as suggested by Yao et al. (2024b).

**Implementation details.** We employ the latent variable identification algorithm proposed by Yao et al. (2024b), which guarantees that the shared latent variables among different views can be identified up to a diffeomorphism in the sense of Defn. B.1. Thus, by utilizing $\mathbf{X}_1, \mathbf{X}_2$, we can obtain a nonlinear bijective transformation of their shared latent variable $\mathbf{Z}_1$. This allows us to construct a measurement model $\mathcal{M} = \langle \mathbf{Z}, \widehat{\mathbf{Z}}_{A_1}, \{h_1\} \rangle$ (see Fig. 2), where $\mathbf{Z} = \{\mathbf{Z}_1, \cdots, \mathbf{Z}_5\}$ and $\widehat{\mathbf{Z}}_{A_1} = h(\mathbf{Z}_1)$ for some (unknown) smooth invertible map $h : \mathbb{R} \to \mathbb{R}$.

We train three CRL models following the implementation settings in (Yao et al., 2024b, Tab. 4).

- **Model A**: a sufficiently trained model (trained for 50001 steps) from which we expect the learned representation $\widehat{\mathbf{Z}}_{A_1}^A$ (where by a slight abuse of notation, the superscript represents the model indicator) to *exclusively measure* $\mathbf{Z}_1$;

- **Model B**: an insufficiently trained model (trained for 51 steps) with unclear latent-measurement correspondence;

- **Model C**: a corrupted version of Model A where the representation $\widehat{\mathbf{Z}}_{A_1}^C \coloneqq \widehat{\mathbf{Z}}_{A_1}^A + 0.2\mathbf{Z}_2 - 0.1\mathbf{Z}_3$, i.e., a linear mixing of the representation $\widehat{\mathbf{Z}}_{A_1}^A$ from Model A, and $\mathbf{Z}_2, \mathbf{Z}_3$.

For each of the three trained models, we generate 50 independent datasets, each containing 4096 paired samples of $\mathbf{Z}, \widehat{\mathbf{Z}}_{A_1}$. We compute the respective T-MEX scores based on these generated datasets for all three models, using the the projected covariance measure (PCM, Lundborg et al., 2024) implemented in `pycomets` (Huang and Kook, 2025) using linear regression models to estimate the conditional means (see App. E).

**Additional results.** Since T-MEX relies on statistical testing, we further assess its statistical validity by examining the underlying p-values that lead to the test results and the T-MEX score. Fig. 6 shows the p-values resulted from testing each of the three null hypotheses:

$$\mathcal{H}_0(i) : \widehat{\mathbf{Z}}_{A_1} \perp\!\!\!\perp \mathbf{Z}_i \,\big|\, \mathbf{Z}_{[5]\setminus i} \text{ for } i \in [3].$$

We omit $\mathbf{Z}_4$ and $\mathbf{Z}_5$ since they are not involved in generating the two views $\mathbf{X}_1$ and $\mathbf{X}_2$.

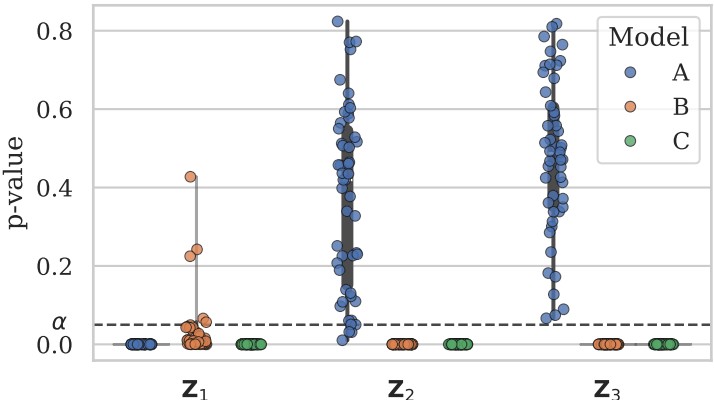

Figure 6: Violin plots of p-values from testing the conditional independencies $\widehat{\mathbf{Z}}_{A_1} \perp\!\!\!\perp \mathbf{Z}_i \mid \mathbf{Z}_{[5]\setminus i}$ for $i \in [3]$ based on the PCM tests (Lundborg et al., 2024). The black dashed line is at the significance level $\alpha = 0.05$. A p-value $< \alpha$ for $\mathbf{Z}_i$ means there is an edge from $\mathbf{Z}_i$ to the measurement $\widehat{\mathbf{Z}}_{A_1}$.

Fig. 6 shows that Model A aligns with the measurement model in Fig. 2, evidenced by (i) small p-values for $\mathcal{H}_0(1)$ and (ii) approximately uniformly distributed p-values for both $\mathcal{H}_0(2)$ and $\mathcal{H}_0(3)$, given a valid test (see App. E for further explanations). In contrast, for Models B and C, nearly all p-values are smaller than $\alpha$, leading to rejections of the null hypotheses, which indicates that the learned representation $\widehat{\mathbf{Z}}_{A_1}$ is a mixture of all three causal variables $\mathbf{Z}_1, \mathbf{Z}_2, \mathbf{Z}_3$, and thus fails to exclusively measure $\mathbf{Z}_1$.

**Computational resources.** We train the CRL models (model A, B, C) using a single node GPU (`NVIDIA GeForce RTX1080Ti`) with 10GB of RAM, 4 CPU cores for less than one GPU hour. ATE estimation and T-MEX computation take less than one minute on a standard CPU.

### D.2 Real-World Ecological Experiment: ISTAnt

**Experiment Setting.** ISTAnt is a real-world ecological benchmark designed to evaluate learned representations on downstream causal inference tasks from high-dimensional observational data. It comprises 44 ant-triplet video recordings collected through a randomized controlled trial. This benchmark adopts the problem formulation introduced by Cadei et al. (2024), aiming to estimate the causal effect of specific treatments (e.g., chemical exposure) on ants' social behavior, particularly grooming events. The experimental design and recording setup are shown in Fig. 7; for further details, refer to (Cadei et al., 2024, App. C).

In ISTAnt, each observation (video recording) $i$ is associated with a treatment assignment $\mathbf{T}_i$ and a set of experimental covariates $\mathbf{W}_i$ (including experiment day, time of the day, batch, position within the batch, and annotator). However, only a subset of videos is annotated with the outcome of interest $\mathbf{Y}_i$ (i.e., grooming events), which hinders reliable causal inference at a population level, such as treatment effect estimation. To address this challenge, Cadei et al. (2024) proposes to train a classifier on top of a pre-trained feature extractor (e.g., DINOv2, Oquab et al., 2023) using this limited set of annotated samples, to impute missing labels while still enabling valid causal inference at the population level; specifically, for estimating the Average Treatment Effect (ATE).

**Implementation details.** Following (Cadei et al., 2024), we train 2,400 classification heads on top of DINOv2 (Oquab et al., 2023), varying the architecture and training settings, and estimate the causal effect using all video samples together with the predicted labels $\widehat{\mathbf{Y}}$s by AIPW estimator (Robins et al., 1994). The hyperparameter configurations are summarized in Tab. 2, with all other implementation details following (Cadei et al., 2024, App. C).

By contrasting with the measurement model depicted in Fig. 4, we compute the T-MEX scores for all 2,400 models. Since we focus on models with more than 80% prediction accuracy (§ 5.2), the null hypothesis $\widehat{\mathbf{Y}} \perp\!\!\!\perp \mathbf{Y} \mid \mathbf{T}$ is rejected in all cases, consistently indicating $\mathbf{Y} \to \widehat{\mathbf{Y}}$. Thus, we only focus

Table 2: Hyperparameters for the real-world ecological experiment (§ 5.2 and App. D.2), giving rise to 2,400 model configurations in total. All other settings follow (Cadei et al., 2024, App. C).

| Hyperparameter | Value(s) |
|---|---|
| Input Preprocessing | YES / NO |
| Number of Hidden Layers | 1, 2 |
| Batch Size | 64, 128, 256 |
| Adam: learning rate | 5e-2, 1e-2, 5e-3, 1e-3, 5e-4 |
| Training objective | Empirical Risk, Invariant Risk (Arjovsky et al., 2020), vREx (Krueger et al., 2021), Deconfounded Risk (Cadei et al., 2025) |
| # Seeds | 0,1, ..., 9 |

on the following null hypothesis:

$$\mathcal{H}_0 : \widehat{\mathbf{Y}} \perp\!\!\!\perp \mathbf{T} \,\big|\, \mathbf{Y},$$

where $\widehat{\mathbf{Y}}$ denotes the predicted label and $\mathbf{Y}$ the ground truth one. A misalignment with the measurement model in Fig. 4 leads to rejecting $\mathcal{H}_0$, resulting T-MEX=1, whereas as a causally valid representation $\widehat{\mathbf{Y}}$ that exclusively measures $\mathbf{Y}$ gives rise to T-MEX=0. We summarize all results in Fig. 5 and provide extended discussions in § 5.2.

**Computational resources.** We run all the analyses in § 5.2 using 48GB of RAM, 20 CPU cores, and a single node GPU (NVIDIA GeForce RTX2080Ti) for 24 GPU hours. Data preprocessing and feature extraction using DINOv2 account for the majority of the computational time, whereas classifier training, AIPW estimation, and the T-MEX test contribute negligibly by comparison.

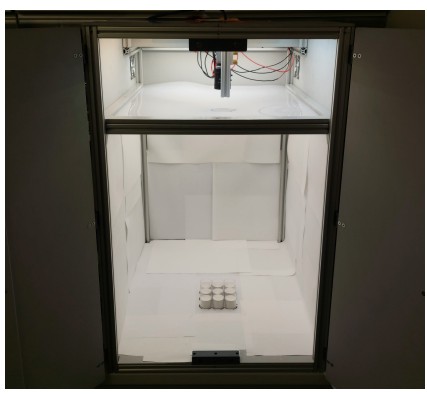

(a) Filming box

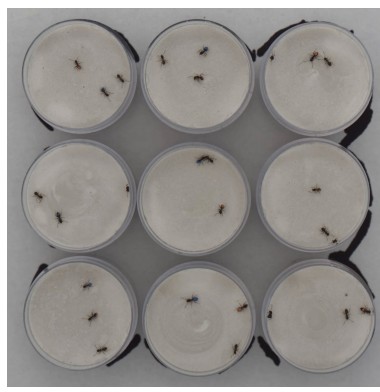

(b) Batch example

Figure 7: Visualization of ISTAnt recording set-up (Cadei et al., 2024).

### D.3 Caveats of Using SHD to Evaluate Causal Representations

**Experiment Setting.** This experiment explores the potential pitfalls when directly using SHD to evaluate causal representations without properly evaluating the element-wise latent variable identification. Specifically, we consider a set of causal variables generated through the following structural equations:

$$\begin{aligned} \mathbf{Z}_1 &= \epsilon_1 \\ \mathbf{Z}_2 &= \alpha_{12} \cdot \mathbf{Z}_1 + \beta_2 \cdot \epsilon_2 \\ \mathbf{Z}_3 &= \alpha_{13} \cdot \mathbf{Z}_1 + \alpha_{23} \cdot \mathbf{Z}_2 + \beta_3 \cdot \epsilon_3, \end{aligned}$$

(D.2)

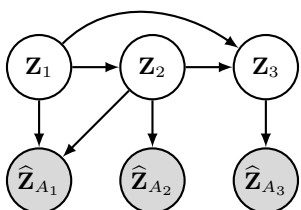

Figure 8: Example measurement model, where $\widehat{\mathbf{Z}}_{A_1}$ block-identifies $\mathbf{Z}_1, \mathbf{Z}_2, \widehat{\mathbf{Z}}_{A_2}$ and $\widehat{\mathbf{Z}}_{A_3}$ identifies $\mathbf{Z}_2, \mathbf{Z}_3$ respectively.

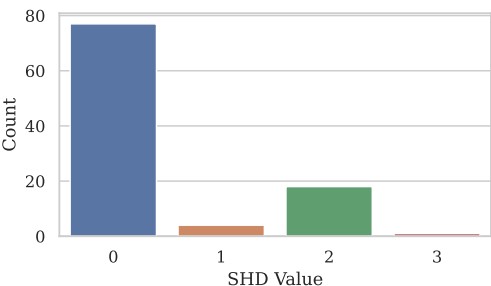

Figure 9: Structural Hamming Distance Values (SHD) of 100 structure and measurement models following eqs. (D.2) and (D.3), where the measurement $\widehat{\mathbf{Z}}_{A_1}$ is a mixing of the ground truth latent $\mathbf{Z}_1, \mathbf{Z}_2$. SHDs are computed between the discovered graph on $\widehat{\mathbf{Z}}$ and the ground truth one.

Assume the learned representation corresponds to the ground truth causal variable as follows:

$$\begin{aligned}
\widehat{\mathbf{Z}}_{A_1} &= \gamma_1 \cdot \mathbf{Z}_1 + \gamma_{21} \cdot \mathbf{Z}_2 \\
\widehat{\mathbf{Z}}_{A_2} &= \gamma_2 \cdot \mathbf{Z}_2 \\
\widehat{\mathbf{Z}}_{A_3} &= \gamma_3 \cdot \mathbf{Z}_3
\end{aligned} \tag{D.3}$$

where $\widehat{\mathbf{Z}}_{A_1}$ remains a mixing of $\mathbf{Z}_1$ and $\mathbf{Z}_2$. The corresponding measurement model is shown in Fig. 8.

**Implementation details.** We generate 100 different structure and measurement models following eqs. (D.2) and (D.3), with all coefficients $\alpha$s and $\gamma$s sampled from $\mathrm{Unif}[1, 10]$ and the $\beta$s sampled from $\mathrm{Unif}[0.005, 0.02]$. We run LiNGAM (Shimizu et al., 2006) from causal-learn (Zheng et al., 2024) to discover the causal relationships between the measurements $\widehat{\mathbf{Z}}_{A_1}, \widehat{\mathbf{Z}}_{A_2}, \widehat{\mathbf{Z}}_{A_3}$.

**Results.** Fig. 9 shows the structural hamming distance of between the discovered graph on $\widehat{\mathbf{Z}}$ and the ground truth one. Despite being entangled between $\mathbf{Z}_1, \mathbf{Z}_2, \widehat{\mathbf{Z}}$ still yield the correct causal graph in most of the cases (77%), as shown by the first bar in the plot. Hence, causal relations between the measurement variables should always be evaluated in conjunction with the variable identification. Otherwise, it can lead to misinterpretations as showcased by Fig. 9.

**Computational resources.** Data generating and causal discovery for App. D.3 in total takes less than 10 minutes on a standard CPU.

### D.4 Additional Numerical Experiments Showcasing the Properties of T-MEX.

In the following, *T-MEX Oracle* is computed based on an oracle CI test with zero type I error and power 1. We compute the oracle T-MEX score to demonstrate desirable properties of the metric—such as its consistency and robustness across different conditional independence tests. This is feasible in the simulated setting, where the measurement functions are known by design. In contrast, for empirical representations learned by a CRL model, the true measurement functions are unknown, making oracle computation infeasible in practice.

**Scalability given higher-dimensional latent variables.** We simulate the causal variables based on a nonlinear SCM – a location-scale SCM as implemented by Wendong et al. (2024). Then, the measurement variables are simulated as a direct copy of each of the corresponding causal variables. For different numbers of latents (n-latent), we report the T-MEX score based on the Generalized Covariance Measure (GCM) test with linear regression (see App. E for more details about GCM), along with their standard error based on 20 random causal DAGs for the latent causal variables, each with 30 repetitions and 1000 observations. In Tab. 3, we see that T-MEX remains closely aligned with the T-MEX oracle in all cases, and can be efficiently computed up to 50 latents within a reasonable time, validating its applicability in moderate to high dimensions.

Table 3: T-MEX for higher-dimensional latents generated from nonlinear SCM

| n-latent | T-MEX | T-MEX Oracle | time (sec) |
|---|---|---|---|
| 5 | $0.0483 \pm 0.2223$ | 0 | $0.0359 \pm 0.0011$ |
| 10 | $0.0033 \pm 0.0577$ | 0 | $0.1559 \pm 0.0017$ |
| 20 | $0.0550 \pm 0.5187$ | 0 | $0.8080 \pm 0.0094$ |
| 50 | $0.0917 \pm 0.9582$ | 0 | $10.8450 \pm 0.1004$ |

**Consistent results when using different CI tests.** Following the same settings in § 5.1, we consider PCM test (see App. E) with random forest (RF) and Kernel Conditional Independence (KCI) test (Zhang et al., 2012). We see that T-MEX ranks the models consistently with the results in § 5.1.

Table 4: T-MEX under different CI Tests

| CI test | Model | T-MEX |
|---|---|---|
| PCM (RF) | A | $0.0000 \pm 0.0000$ |
| PCM (RF) | B | $0.8000 \pm 0.7559$ |
| PCM (RF) | C | $2.0000 \pm 0.0000$ |
| KCI | A | $0.0400 \pm 0.1979$ |
| KCI | B | $0.1600 \pm 0.4218$ |
| KCI | C | $2.0000 \pm 0.0000$ |

**T-MEX under weak causal relations.** Building on the experiment in § 5.1, we next examine how T-MEX behaves under weak causal relations, and compare it with the previous CRL identifiability metrics such as $R^2$ and MCC. The data is generated from a linear SCM with three latent causal variables following the measurement model described in Fig. 9. The linear coefficients between the causal variables are sampled uniformly between 0.01 and 0.1.

Table 5: T-MEX under weak causal relations

| T-MEX Oracle | T-MEX | $R^2$ | MCC |
|---|---|---|---|
| 1 | $1.0100 \pm 0.0995$ | $1.0000 \pm 0.0000$ | $1.0000 \pm 0.0000$ |

Under identifiability assumptions that guarantee element-wise correspondence, T-MEX correctly detects the mixing effect and gives a score near one (note that T-MEX = 0 indicates perfect alignment). In contrast, both $R^2$ and MCC fail to reflect this violation. Despite the entanglement in $\widehat{\mathbf{Z}}_1$, they still assign the maximum score of 1, misleadingly suggesting perfect element-wise identification (see Tab. 5).

**Alignment between T-MEX and ATE under nonlinear SCM.** Extending § 5.1, we further examine the correspondence between T-MEX and ATE under the more general nonlinear setting. We consider the same causal graph as given in § 5.1, where $\mathbf{Z}_1$ confounds $\mathbf{Z}_2$ and $\mathbf{Z}_3$, and needs to be adjusted for a valid treatment effect estimation. A "perfect" model means $\widehat{\mathbf{Z}}_1$ exclusively measures $\mathbf{Z}_1$, whereas a mixed model indicates an entangled representation, i.e., $\widehat{\mathbf{Z}}_1$ mixes $\mathbf{Z}_1$ and $\mathbf{Z}_2$.

Table 6: T-MEX v.s. ATE bias under nonlinear SCM

| Model | T-MEX | abs(ATE bias) |
|---|---|---|
| Perfect | $0.0100 \pm 0.1000$ | $0.1532 \pm 0.0283$ |
| Mixed | $1.0000 \pm 0.0000$ | $0.5357 \pm 0.0540$ |

As shown in Tab. 6, the results are highly similar to the linear case § 5.1, T-MEX closely aligns with the absolute values of the ATE bias, effectively evaluating causal representations for downstream inference tasks with nonlinear causal relations.

**Consistency of T-MEX under noisy measurements.** We compare the empirical T-MEX value with/without noise. The noisy measurements are generated by adding Gaussian noise to the original latents. I.e., the measurement function writes

$$h(\mathbf{z}) = \mathbf{z} + e$$

with $e$ independent Gaussian noise. Results are evaluated over 100 datasets with 1000 samples each. Tab. 7 shows that the empirical T-MEX score largely remains consistent under noisy measurements, validating its practical usability.

Table 7: T-MEX w/wo noise

| T-MEX Oracle | without noise | with noise |
|:---:|:---:|:---:|
| 0 | $0.0100 \pm 0.0100$ | $0.3900 \pm 0.6497$ |

# E   Background on Conditional Independence Testing

Testing conditional independence of two random variables $\mathbf{X}$ and $\mathbf{Y}$ given a third random variable $\mathbf{Z}$ is known to be a difficult problem if $Z$ is a continuous variable (Shah and Peters, 2020). The goal of conditional independence test is to test the null hypothesis

$$\mathcal{H}_0 : \mathbf{X} \perp\!\!\!\perp \mathbf{Y} \mid \mathbf{Z}.$$

Shah and Peters (2020) have shown that there is no valid test (i.e., a test that guarantees a Type I error rate to be no larger than the given significance level $\alpha$) that has power against all alternatives.

Consider univariate variables $\mathbf{X}, \mathbf{Y}, \mathbf{Z}$, the generalized covariance measure (GCM) test proposed in Shah and Peters (2020) aims to test an implication of conditional independence which can be written as the following null hypothesis:

$$\mathcal{H}_0^{\mathrm{GCM}} : \mathbb{E}[(\mathbf{Y} - \mathbb{E}[\mathbf{Y} \mid \mathbf{Z}])(\mathbf{X} - \mathbb{E}[\mathbf{X} \mid \mathbf{Z}])] = 0.$$

The validity of the GCM test thus relies on that the conditional means $\mathbb{E}[\mathbf{Y} \mid \mathbf{Z}]$ and $\mathbb{E}[\mathbf{X} \mid \mathbf{Z}]$ can be learned at sufficiently fast rates. It turns out that GCM does not have power against any alternative for which $\mathbb{E}[\mathrm{Cov}(\mathbf{X}, \mathbf{Y} \mid \mathbf{Z})] = 0$ but $\mathbf{X} \not\perp\!\!\!\perp \mathbf{Y} \mid \mathbf{Z}$ (Lundborg et al., 2024).

The projected covariance measure (PCM) proposed by Lundborg et al. (2024) improves the power issue of GCM by testing a different implication of conditional independence:

$$\mathcal{H}_0^{\mathrm{PCM}} : \mathbb{E}[\mathbf{Y} \mid \mathbf{X}, \mathbf{Z}] = \mathbb{E}[\mathbf{Y} \mid \mathbf{Z}].$$

Similar to GCM, to ensure its validity, PCM also requires that the conditional means can be learned sufficiently fast, which is satisfied in our experiments (§ 5).

There are other conditional independence tests such as mutual information based methods (Ai et al., 2024; Runge, 2018) and kernel-based methods (Fernández and Rivera, 2024; Strobl et al., 2019; Zhang et al., 2012). We opted for PCM in our experiments for its computational advantage and theoretical guarantees on its validity under a flexible, model-agnostic framework. More discussions on the usage of PCM and GCM can be found in Kook and Lundborg (2024). Notably, T-MEX is a general evaluation metric for causal representations that does not specify any particular type of tests, allowing practitioners to choose other testing methods that are more suitable for their problem settings.

# F   Extended Discussion

This section elaborates on the implications of learned representations for downstream causal tasks. As briefly discussed in the main paper (following Defn. 2.2), a representation is *causally valid* (Defn. 2.2) with respect to a statistical estimand if and only if the statistical estimand remains unchanged when

plugging in the measurement variables correspond to the causal variables. More concretely, we illustrate the implications of nonlinear invertible reparameterizations of causal variables in two commonly encountered scenarios: when representations serve as proxies of (i) the treatment or outcome variables, and (ii) the confounders or instrumental variables.

### F.1 Representations of Treatment and Outcome

Assume in Fig. 10 that $\widehat{\mathbf{Z}}_{A_1}, \widehat{\mathbf{Z}}_{A_2}$ are element-wise non-linear invertible reparametrization of $\mathbf{Z}_1, \mathbf{Z}_2$ respectively; i.e., $\forall i \in \{1, 2\}, \widehat{\mathbf{Z}}_{A_i} = h_i(\mathbf{Z}_i)$ for some diffeomorphism $h_i : \mathbb{R} \to \mathbb{R}$. We aim to estimate the treatment effect of $\mathbf{Z}_1 \to \mathbf{Z}_2$ using the learned representations $\widehat{\mathbf{Z}}_{A_1}$ and $\widehat{\mathbf{Z}}_{A_2}$.

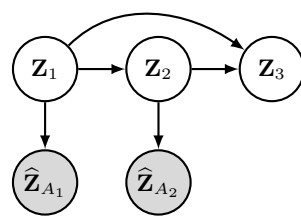

Assume the $\mathbf{Z}_2$ is generated following eq. (4.1), i.e.,

$$\mathbf{Z}_2 \coloneqq a \cdot \mathbf{Z}_1 + e$$

with $e \sim P_e$, $\mathbb{E}[e] = 0$ and $e \perp\!\!\!\perp \mathbf{Z}_1$. Given there is no un-observed confounding, the ground truth average treatment effect is written as

Figure 10: $\widehat{\mathbf{Z}}_{A_i}$ measures $\mathbf{Z}_i$ through a nonlinear bijection for both $i = 1, 2$.

$$\text{ATE}(\mathbf{Z}_1 \to \mathbf{Z}_2) = \frac{\partial \mathbb{E}[\mathbf{Z}_2 \mid do(\mathbf{Z}_1 = \mathbf{z}_1)]}{\partial \mathbf{z}_1} = \frac{\partial \mathbb{E}[\mathbf{Z}_2 \mid \mathbf{Z}_1 = \mathbf{z}_1]}{\partial \mathbf{z}_1} = \frac{\partial \mathbb{E}[a\mathbf{z}_1 + e]}{\partial \mathbf{z}_1} = a. \quad \text{(F.1)}$$

We assume measurement function $h_i$ for all $i \in \{1, 2\}$ to be linear, i.e.,

$$\widehat{\mathbf{Z}}_{A_1} = \alpha_1 \cdot \mathbf{Z}_1, \qquad \widehat{\mathbf{Z}}_{A_2} = \alpha_2 \cdot \mathbf{Z}_2, \quad \text{and} \quad \alpha_1, \alpha_2 \neq 0. \quad \text{(F.2)}$$

The ATE estimand from the learned representations yields:

$$\begin{aligned}
\text{ATE}(\widehat{\mathbf{Z}}_{A_1} \to \widehat{\mathbf{Z}}_{A_2}) &= \frac{\partial \mathbb{E}[\widehat{\mathbf{Z}}_{A_2} \mid \widehat{\mathbf{Z}}_{A_1} = \hat{\mathbf{z}}_{A_1}]}{\partial \hat{\mathbf{z}}_{A_1}} \\
&= \frac{\partial \mathbb{E}[\alpha_2 \mathbf{Z}_2 \mid \alpha_1 \mathbf{Z}_1 = \alpha_1 \mathbf{z}_1]}{\partial \alpha_1 \mathbf{z}_1} \\
&= \frac{\alpha_2 \partial \mathbb{E}[\mathbf{Z}_2 \mid \mathbf{Z}_1 = \mathbf{z}_1]}{\alpha_1 \partial \mathbf{z}_1} = \frac{\alpha_2}{\alpha_1} a.
\end{aligned} \quad \text{(F.3)}$$

As shown by eq. (F.3), the ATE estimand using the learned representation $\widehat{\mathbf{Z}}_{A_1}$ and $\widehat{\mathbf{Z}}_{A_2}$ can be arbitrarily scaled by the factor of $\alpha_2/\alpha_1$. Thus, measurements that bijectively transform the causal latent variables cannot naively support estimating the treatment effect, violating causal validity (Defn. 2.2); it requires direct supervision or observation on *both* treatment and outcome variables, as also pointed out by (von Kügelgen et al., 2024, Sec. 4).

On the other hand, information-theoretic measures for quantifying causal influence remain invariant under bijective transformation, such as the mutual information $I_{\text{int}}(\mathbf{Z}_1; \mathbf{Z}_2) = I_{\text{int}}(\widehat{\mathbf{Z}}_{A_1}; \widehat{\mathbf{Z}}_{A_2})$, as shown by Janzing et al. (2013).

### F.2 Representations of Confounders or Instruments

**Measuring confounding.** We first show an example where an observed treatment $\mathbf{T}$ and an observed outcome $\mathbf{Y}$ is confounded by a third variable $\mathbf{W}$ which is measured by $\widehat{\mathbf{W}} = h(\mathbf{W})$ through a deterministic invertible function $h$.

Formally, the measurement model is defined as $\mathcal{M}^{\text{conf}} = \langle \mathbf{Z}, \widehat{\mathbf{Z}}, \{h\} \rangle$ with $\mathbf{Z} = \{\mathbf{T}, \mathbf{Y}, \mathbf{W}\}$ and $\widehat{\mathbf{Z}} = \{\widehat{\mathbf{W}}\}$, where $\mathbf{T}, \mathbf{Y}$ are *directly measured* (Defn. 2.1). The corresponding DAG is given in Fig. 11. We show in the following that this measurement model is indeed causally valid (Defn. 2.2) with respect to the statistical estimand for the Average Treatment Effect (ATE) of $\mathbf{T}$ on $\mathbf{Y}$.

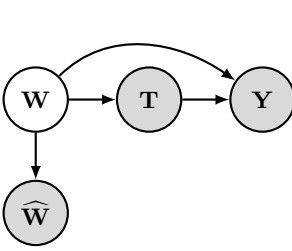

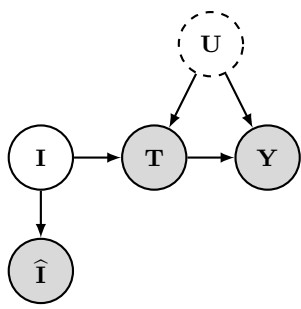

Figure 11: *ATE remains invariant under bijective transformation of confounders.* The treatment $\mathbf{T}$ and outcome $\mathbf{Y}$ are directly measured (i.e., observed) whereas confounder $\mathbf{W}$ is measured by $\widehat{\mathbf{W}}$ through a nonlinear bijection.

Figure 12: *ATE remains invariant under bijective transformation of instruments.* $\widehat{\mathbf{I}}$ measures the instrument variable $\mathbf{I}$ through a nonlinear bijection. The treatment $\mathbf{T}$ and outcome $\mathbf{Y}$ are directly measured (i.e., observed), and $\mathbf{U}$ denotes unobserved confounding.

Under the standard assumptions for backdoor adjustment, it follows that

$$
\begin{aligned}
\mathbb{E}(\mathbf{Y}|do(\mathbf{T}=t)) &= \mathbb{E}_{\mathbf{w}}\left[\mathbb{E}(\mathbf{Y}\mid\mathbf{W},\mathbf{T}=t)\right] \\
&= \int \mathbb{E}(\mathbf{Y}\mid\mathbf{W},\mathbf{T}=t)P(\mathbf{W})d\mathbf{w} \\
&= \int \mathbb{E}(\mathbf{Y}\mid h^{-1}(\widehat{\mathbf{W}}),\mathbf{T}=t)P(h^{-1}(\widehat{\mathbf{W}}))\frac{dh^{-1}(\hat{\mathbf{w}})}{d\hat{\mathbf{w}}}d\hat{\mathbf{w}} \\
&= \int \mathbb{E}(\mathbf{Y}\mid\widehat{\mathbf{W}},\mathbf{T}=t)P(\widehat{\mathbf{W}})d\hat{\mathbf{w}} \\
&= \mathbb{E}_{\hat{\mathbf{w}}}\left[\mathbb{E}(\mathbf{Y}\mid\widehat{\mathbf{W}},\mathbf{T}=t)\right],
\end{aligned}
\tag{F.4}
$$

where we used the change of variable formula and the fact that $\mathbb{E}(\mathbf{Y}\mid\widehat{\mathbf{W}},\mathbf{T}=t) = \mathbb{E}(\mathbf{Y}|h^{-1}(\widehat{\mathbf{W}}),\mathbf{T}=t)$. This is because $h^{-1}(\widehat{\mathbf{W}})$ is a sufficient statistic for $\mathbf{W}$ (Casella and Berger, 2024, Ch. 6.2) following $h$ is invertible.

Under the same assumptions, the ATE for *binary* treatment can then be identified by the following statistical estimand

$$
\begin{aligned}
\text{ATE}(\mathbf{T}\to\mathbf{Y}) &= \mathbb{E}[\mathbf{Y}|do(\mathbf{T}=1)] - \mathbb{E}[\mathbf{Y}\mid do(\mathbf{T}=0)] \\
&= \mathbb{E}_{\mathbf{w}}\left[\mathbb{E}(\mathbf{Y}\mid\mathbf{W},\mathbf{T}=1) - \mathbb{E}(\mathbf{Y}\mid\mathbf{W},\mathbf{T}=0)\right].
\end{aligned}
\tag{F.5}
$$

Following eq. (F.4), we have

$$
\text{ATE}(\mathbf{T}\to\mathbf{Y}) = \mathbb{E}_{\hat{\mathbf{w}}}\left[\mathbb{E}(\mathbf{Y}\mid\widehat{\mathbf{W}},\mathbf{T}=1) - \mathbb{E}(\mathbf{Y}\mid\widehat{\mathbf{W}},\mathbf{T}=0)\right],
$$

indicating that the identified statistical estimand $\text{ATE}(\mathbf{T}\to\mathbf{Y})$ remains invariant for the measurement $\widehat{\mathbf{W}}$. Similarly, ATE also remains invariant when the treatment is continuous:

$$
\text{ATE}(\mathbf{T}\to\mathbf{Y}) = \frac{\partial\mathbb{E}[\mathbf{Y}\mid do(\mathbf{T}=t)]}{dt} = \frac{\partial\mathbb{E}_{\mathbf{w}}\mathbb{E}[\mathbf{Y}\mid\mathbf{W},\mathbf{T}=t]}{dt} = \frac{\partial\mathbb{E}_{\hat{\mathbf{w}}}\mathbb{E}[\mathbf{Y}\mid\widehat{\mathbf{W}},\mathbf{T}=t]}{dt},
\tag{F.6}
$$

where the last equality holds because of eq. (F.4). Therefore, we have shown that invertible reparameterizations of the confounders can be a drop-in replacement of the true confounding variables in the statistical estimand for ATE, for both discrete and continuous treatments, and thus this measurement model $\mathcal{M}^{\text{conf}}$ is indeed causally valid for ATE.

**Measuring instrumental variables.** We now give a second example of ATE estimation under an instrumental variable setup. We assume that the instrument $\mathbf{I}$ is measured by $\widehat{\mathbf{I}} = h(\mathbf{I})$ through a bijective transformation $h$. We show that under certain assumptions, the statistical estimand does not change when using $\widehat{\mathbf{I}}$ as a drop-in replacement of the true instrument $\mathbf{I}$. We focus on the case where the

instrument $\mathbf{I}$, the treatment $\mathbf{T}$, and the response $\mathbf{Y}$ are all univariate continuous variables; further discussion on multivariate and discrete valued variables is beyond the scope of this paper. Formally, the measurement model is defined as $\mathcal{M}^{\text{IV}} = \langle \mathbf{Z}, \widehat{\mathbf{Z}}, \{h\} \rangle$ with causal variables $\mathbf{Z} = \{\mathbf{I}, \mathbf{T}, \mathbf{Y}\}$ and measurement variables $\widehat{\mathbf{Z}} = \{\widehat{\mathbf{I}}\}$. The treatment $\mathbf{T}$ and outcome $\mathbf{Y}$ are *directly measured* (Defn. 2.1) and confounded by unknown hidden confounders $\mathbf{U}$. Fig. 12 shows the DAG of this measurement model.

We show in the following that the instrument $\mathbf{I}$ remains a valid instrumental variable under a bijective transformation, i.e., the measurement variable $\widehat{\mathbf{I}} = h(\mathbf{I})$ also satisfies the standard IV assumptions, which are listed as follows:

- Relevancy: $\mathbf{I} \not\perp\!\!\!\perp \mathbf{T} \mid \mathbf{U}$
- Unconfoundedness: $\mathbf{I} \perp\!\!\!\perp \mathbf{U}$
- Exclusion restriction criteria: $\mathbf{I} \perp\!\!\!\perp \mathbf{Y} \mid \mathbf{T}, \mathbf{U}$

Following standard probability theory (see e.g., Billingsley, 2008), if $h$ is a bijective function, all three conditions still hold when replacing $\mathbf{I}$ by $h(\mathbf{I})$. This means that if the ATE is identified by a statistical estimand when using $\mathbf{I}$ as an instrument, it is also identified when using $\widehat{\mathbf{I}}$ as an instrument. In other words, the measurement model $\mathcal{M}^{\text{IV}}$ is causally valid with respect to an identified statistical estimand because $\widehat{\mathbf{I}}$ can serve as a drop-in replacement for $\mathbf{I}$ (Defn. 2.2).

As a specific example, consider the case where the causal mechanism of $\mathbf{Y}$ is partially linear (a commonly studied setup in the semi-parametric inference literature, see e.g., Chernozhukov et al. (2018)), i.e., $\mathbf{Y} = \mathbf{T}\beta + g(\mathbf{U}, \varepsilon)$, for some measurable function $g$ where $\mathbb{E}[g(\mathbf{U}, \epsilon)] = 0$ and where $\varepsilon \sim P_\varepsilon$ is an independent noise variable, the ATE

$$\text{ATE}(\mathbf{T} \rightarrow \mathbf{Y}) = \frac{\partial \mathbb{E}[\mathbf{Y} \mid do(\mathbf{T} = \mathbf{t})]}{\partial \mathbf{t}} = \frac{\partial \mathbb{E}[\mathbf{t}\beta + g(\mathbf{U}, \varepsilon)]}{\partial \mathbf{t}} = \beta$$

can be identified by the statistical estimand

$$\text{ATE}(\mathbf{T} \rightarrow \mathbf{Y}) = \frac{\text{Cov}(\mathbf{Y}, \mathbf{I})}{\text{Cov}(\mathbf{T}, \mathbf{I})}. \tag{F.7}$$

We show in the following that the statistical estimand $\text{ATE}(\mathbf{T} \rightarrow \mathbf{Y})$ in eq. (F.7) remains invariant when using $\widehat{\mathbf{I}}$ as a drop-in replacement for $\mathbf{I}$. Plugging in $\widehat{\mathbf{I}}$ in the numerator

$$\text{Cov}(\mathbf{Y}, \widehat{\mathbf{I}}) = \mathbb{E}[\mathbf{Y}\widehat{\mathbf{I}}] - \mathbb{E}[\mathbf{Y}]\mathbb{E}[\widehat{\mathbf{I}}] = \beta \left( \mathbb{E}[\mathbf{T}\widehat{\mathbf{I}}] - \mathbb{E}[\mathbf{T}]\mathbb{E}[\widehat{\mathbf{I}}] \right) = \beta \text{Cov}(\mathbf{T}, \widehat{\mathbf{I}}),$$

we have $\dfrac{\text{Cov}(\mathbf{Y}, \widehat{\mathbf{I}})}{\text{Cov}(\mathbf{T}, \widehat{\mathbf{I}})} = \beta = \dfrac{\text{Cov}(\mathbf{Y}, \mathbf{I})}{\text{Cov}(\mathbf{T}, \mathbf{I})}$. Therefore, we have shown another example where the measurement $\widehat{\mathbf{I}}$ can serve as a drop-in replacement for the latent instrumental variable $\mathbf{I}$ for downstream causal inference tasks.

