# OpenReview forum: "The third pillar of causal analysis? A measurement perspective on causal representations"
_NeurIPS.cc/2025/Conference — NeurIPS 2025 poster_

### Official Review · Reviewer_yCFv · 2025-06-13

**Clarity:** 2
**Significance:** 2
**Originality:** 2
**Rating:** 4
**Confidence:** 3

**Summary:**

The authors aim to clarify the role of CRL in causal analysis, particularly when the causal variables are not directly observable. The paper introduces a new evaluation metric, the Test-based Measurement EXclusivity (T-MEX) score, to assess how well learned representations align with the underlying causal structure. The T-MEX score is validated through both theoretical analysis and empirical experiments in numerical simulations and ecological video analysis.

**Questions:**

1. How well does this method generalize to more complex causal environments with multiple hidden variables and interactions? Could you provide examples of this and does T-MEX scale efficiently to high-dimensional datasets or when the number of causal variables increases significantly? Are there any known limitations in scaling the methodology to large datasets or real-time applications?

2. How does T-MEX perform in comparison to existing evaluation metrics in causal representation learning in cases where causal dependencies are weak or non-linear?

3. You discuss extensions to noisy data in your measurement model framework. Can you elaborate on how T-MEX behaves when noise levels are high or when data is missing?

**Ethical Concerns:**

["NO or VERY MINOR ethics concerns only"]

**Final Justification:**

Most of all my concerns are resolved.

**Limitations:**

Yes

**Quality:**

2

**Strengths And Weaknesses:**

**Strengths**
1. The T-MEX score is a unique and practical contribution, offering a new way to quantify the alignment between learned representations and their causal counterparts. This metric is applied effectively in both simulated and real-world settings.

2. The paper thoroughly addresses the existing challenges in causal analysis and representation learning, proposing a novel and interpretable solution.


**Weaknesses**

1. While the framework and T-MEX score are valuable, they may be challenging for practitioners unfamiliar with the underlying mathematical foundations or causal inference techniques. The paper could benefit from clearer explanations or visual aids for those new to the field.

2. The paper discusses its framework’s ability to generalize across various causal tasks, but it remains to be seen how well it handles more complex or diverse causal environments beyond the tested scenarios.

---

> ### Author Rebuttal · Authors · 2025-07-30
>
> Thank you for carefully assessing our work and for considering our paper as “*a unique and practical contribution”*. We provide detailed responses to the questions as follows.
>
> &nbsp;
>
> ## Additional Experimental Results
>
> **T-MEX Oracle:** T-MEX Oracle is computed based on an oracle CI test with 0 type I error and power 1. We compute the oracle T-MEX score to demonstrate desirable properties of the metric—such as its consistency and robustness across different conditional independence tests. This is feasible in the simulated setting, where the measurement functions are known by design. In contrast, for empirical representations learned by a CRL model, the true measurement functions are unknown, making oracle computation infeasible in practice.
>
> **Table 1: T-MEX for higher-dimensional latents generated from nonlinear SCM**
>
> | n\_latent | T-MEX                        | T-MEX Oracle |  time  (sec)   |
> |:-------------------:|:----------------------------:|:------------:| -----------------------:|
> | $5$                 | $0.0483 \pm 0.2223$          | $0$          | $0.0359 \pm 0.0011$    |
> | $10$                | $0.0033 \pm 0.0577$          | $0$          | $0.1559 \pm 0.0017$    |
> | $20$                | $0.0550 \pm 0.5187$          | $0$          | $0.8080 \pm 0.0094$    |
> | $50$                | $0.0917 \pm 0.9582$          | $0$          | $10.8450 \pm 0.1004$    |
>
>
> * **Experimental Setup:** We simulate the causal variables based on a nonlinear SCM – a location-scale SCM as implemented by **`Liang et. al. 2023`**. Then, the measurement variables are simulated as a direct copy of each of the corresponding causal variables. For different numbers of latents (n\_latent), we report the T-MEX score based on the Generalized Covariance Measure (GCM) test with linear regression (see **`Appendix E`** for more details about GCM), along with their standard error based on 20 random causal DAGs for the latent causal variables, each with 30 repetitions and 1000 observations.
> * **Results:** T-MEX remains closely aligned with the T-MEX oracle in all cases, and can be efficiently computed up to 50 latents within reasonable time, validating its applicability in moderate to high dimensions.
>
> *Liang, W., Kekić, A., von Kügelgen, J., Buchholz, S., Besserve, M., Gresele, L., & Schölkopf, B. (2023). Causal component analysis. Advances in Neural Information Processing Systems, 36, 32481-32520.*
> *Reference*
>
> **The data-generating process for Table 5** follows the example in **`Appendix D.3`** with minor modifications. Here we provide a brief introduction of the example for convenience:
>
> * This example consists of three causal variables \\( \\mathbf{Z}\_1,\\mathbf{Z}\_2,\\mathbf{Z}\_3 \\) and we obtain three entangled measurements  \\( \\widehat{\\mathbf{Z}}\_1, \\widehat{\\mathbf{Z}}\_2 \\) and \\( \\widehat{\\mathbf{Z}}\_3 \\)  for the latent causal variables.
> * Specifically, \\( \\widehat{\\mathbf{Z}}\_2 \\) and \\( \\widehat{\\mathbf{Z}}\_3 \\) directly correspond to \\( \\mathbf{Z}\_2 \\) and \\( \\mathbf{Z}\_3 \\), while \\( \\widehat{\\mathbf{Z}}\_1 \\) remains a ***mixture*** of \\( \\mathbf{Z}\_1 \\) and \\( \\mathbf{Z}\_2 \\)*,* as shown in **`Figure 8 in Appendix D.3`**.
> * We assume the element-wise identification as the ground-truth measurement model (a common identification guarantee by many CRL methods). In this case, the T-MEX Oracle yields one.
>
> **General clarification for Mean $R^2$ and MCC computation**: Both Mean $R^2$ and MCC are computed under the best index assignment, i.e, the permutation that gives the highest score, following the standard implementation in CRL.
>
> **Table 5:  T-MEX, $R^2$, and MCC under weak or nonlinear causal relations**
>
> |Causal Relations| T-MEX Oracle |   T-MEX  |       Mean $R^2$      |   MCC   |
> |:-------------:|:-------------:|:---------:|:-------------------:|:--------:|
> | weak | $1$          |$1.0100 \\pm 0.0995$|$1.0000\\pm 0.0000$|$1.0000 \\pm 0.0000$|
> | nonlinear| $1$          |$1.0000 \\pm 0.0000$|$0.9711 \\pm 0.0015$|$0.9852 \\pm 0.0008$|
>
> * **Experimental Setup:**
>   * **Weak**: We generate data from a linear SCM with three latent causal variables following the DGP described above. **The linear coefficients between the causal variables are sampled uniformly between 0.01 and 0.1.**
>   * **Nonlinear**: We generate data from a **partially non-linear SCM** with three latent causal variables following the DGP described above. The causal relationship between $\\mathbf{Z}\_1$ and $\\mathbf{Z}\_2$ as well as between $\\mathbf{Z}\_1$ and $\\mathbf{Z}\_3$ is nonlinear (a leaky tanh function) with additive noise, while the causal relationship between $\\mathbf{Z}\_2$ and $\\mathbf{Z}\_3$ is linear.
>   * In both cases, we repeated the experiments 100 times each with 1000 observations.
> * **Results:** In both cases, T-MEX yields a score of 1 (up to some variance), indicating exclusivity violation in $\\widehat{\\mathbf{Z}}\_1$, whereas Mean $R^2$ and MCC provide misleading results (with the highest score of 1\) that suggest perfect element-wise identification. See **`Q2`** for a more detailed explanation.
>
> **Table 6: Consistency of T-MEX under noisy measurements**
>
> |T-MEX Oracle|      witout noise     |      with noise   |
> |:------------:|:------------------:|:------------------:|
> |      0     |$0.0100 \\pm 0.1000$| $0.3900 \\pm 0.6497$|
>
> * **Experimental Setup:** We compare the empirical T-MEX value with/without noise. The noisy measurements are generated by adding Gaussian noise to the original latents. I.e., the measurement function writes $h(z) \= z \+ e$ with $e$ independent Gaussian noise. Results are evaluated over 100 datasets with 1000 samples each.
> * **Results:** T-MEX remains largely consistent under noisy measurements. See **`Q3`** for a more detailed explanation.
>
> &nbsp;
>
> ## Response to Weaknesses
>
> ***W1: More explanations or visual aids***
> Great point—thank you\! We will add a detailed explanation of the framework along with step-by-step usage guidelines in the appendix. Additionally, we will provide a modular and user-friendly implementation in the code library that can be used as a black-box.
>
> ***W2: More complex or diverse causal environments***
> Thanks for your suggestion\! We have provided additional experimental results on **higher-dimensional latents (`Table 1`),**  and for latents under **weak or nonlinear causal relations (`Table 5`).** We also demonstrate T-MEX’s effectiveness under noisy measurements, as shown in **`Table 6`**.
>
> &nbsp;
>
> ## Response to Questions
>
> ***Q1: Scale up the experiments and known limitations in CI tests***
>
> - **Scale up the experiments:** We have included additional T-MEX results in **`Table 1`** for different numbers of latents (up to 50) generated from a nonlinear structural causal model (SCM). We also report the runtime of T-MEX computation as a function of the number of latent variables, demonstrating its practical feasibility even in moderately high-dimensional settings.
>
>
>
> - **Known scaling limitations of CI tests:**  It is well known that the statistical power of conditional independence (CI) tests deteriorates as the dimensionality of the conditioning set increases—a limitation shared by most CI-based causal discovery methods **`[Zhang et al., 2012; Shah & Peters, 2020; Strobl et al., 2019]`**. Since our proposed framework is agnostic to CI test choices, T-MEX can be easily scaled to larger dimensions by adapting more powerful and scalable test methods as they are developed.
>
>
> *Zhang, K., Peters, J., Janzing, D., & Schölkopf, B. (2012). Kernel-based conditional independence test and application in causal discovery. UAI.*
>
> *Shah, R. D., & Peters, J. (2020). The hardness of conditional independence testing and the generalised covariance measure. Annals of Statistics.*
>
> *Strobl, E. V., Zhang, K., & Visweswaran, S. (2019). Approximate kernel-based conditional independence tests for fast non-parametric causal discovery. Journal of Causal Inference.*
>
> ***Q2: Comparison to existing evaluation metrics in CRL under weak or nonlinear causal relations***
>
> * **`Table 5`** compares T-MEX with $R^2$ and MCC for latents generated from **weak and nonlinear** causal relations.
> * Under identifiability assumptions that guarantee element-wise correspondence, T-MEX correctly detects the mixing effect in $\\widehat{\\mathbf{Z}}\_1$ and gives a score near one (note that T-MEX \= 0 indicates perfect alignment).
> * In contrast, both mean $R^2$ and MCC fail to reflect this violation. Despite the entanglement in $\\widehat{\\mathbf{Z}}\_1$​, they still assign the maximum score of 1, misleadingly suggesting perfect element-wise identification.
>
>
>
> ***Q3: T-MEX under noise variables***
>
> We clarify below how T-MEX behaves under high noise levels and in the presence of missing data:
>
> * **High noise levels:** As discussed in **`Remark 2.2`**, noise in the measurement functions can be treated as **additional independent latent variables**. This means that if the noise is independent of the existing causal variables, the **conditional independencies required for T-MEX remain intact**. In such cases, one can safely compute T-MEX without explicitly modeling the noise; the score will remain valid, as shown in **`Table 6`.**
>
> * **Missing data:** Great suggestion\! If we know which nodes block the paths between the variables we need to test, then T-MEX can also handle the missing labels of certain ground truth causal variables, as long as the path they appear in is otherwise blocked.

---

> > ### Comment · Reviewer_yCFv · 2025-08-05
> >
> > I appreciate the author's response. I am pleased to see experimental results addressing high-dimensional representations and noise. Accordingly, I will raise my score to 4.

---

> > > ### Author Response · Authors · 2025-08-07
> > >
> > > Thank you for your time and effort in reviewing our paper and responses. We sincerely appreciate your thoughtful feedback and constructive suggestions. We're glad to hear that you're willing to raise your score, and we will ensure that the points discussed are carefully incorporated into our revision.

---

### Official Review · Reviewer_MeWB · 2025-06-27

**Clarity:** 3
**Significance:** 4
**Originality:** 3
**Rating:** 5
**Confidence:** 4

**Summary:**

The authors propose T-MEX, a new score to evaluate the quality of learned causal representations. Compared to existing metrics such as $R^2$ and MCC, the authors show that T-MEX is more adequate to assess how good the representation will be for causal downstream tasks. The idea behind this metric, based on conditional independence tests, is to measure the difference between 1) the graph linking the ground-truth and the learned representations and 2) the assumed graph between the ground-truth and the measurement model. Finally, the authors validate their score on synthetic and real-world data, demonstrating that a lower value of T-MEX is indeed associated with a smaller average treatment effect bias.

**Questions:**

- While I understand the usefulness of the measurement model when considering models learning block-identifiable representation and specific causal tasks, the usefulness of the metric is less clear to me when considering models with identifiable representation (up to permutation & scaling), which is the most common case. In this scenario, the measurement model is simply mapping each variable to its equivalent measurement.

- Besides the comment about the inadequacy of MCC, no experiments on the paper show this case and how MCC performs compared to T-MEX. I’d be interested to see this comparison since this is the most common case in CRL.

- Considering the same scenario, T-MEX seems like a crude score: since T-MEX assesses in a discrete manner the existence/absence of conditional independence, one could have the same T-MEX score for two highly different representations that lead to different results on downstream tasks. One that strongly correlates with the ground-truth latent, and another one that weakly correlates, but is enough to be dependent, according to the CI test used.

- For the experiment in Section 5.1, while it is said that $R^2$ fails to show a clear correspondence to the ATE bias, the average $R^2$ leads to the same ordering of the model as T-MEX.

- Finally, while I recommend the authors for using a real-world dataset to validate their method, I find this experiment limited. Since there is only one measure variable and one ground-truth latent, the only two possible graphs are the empty graph or $Y$ causes $\hat{Y}$. The empty graph means that the learned representation is independent of the ground-truth which seems like an overall bad model. Despite this, the differences observed in Figure 5 are pretty modest.

__Typos:__
- Line 121: diffeomporphism
- Line 265: the MCC would obtain the highest value 1 although…
- V is both used to denote the variables and the graph between them. I would use a different notation.
- In Section 5.1, both sd and std are used to represent standard deviation. I would choose one notation and stick to it.

**Ethical Concerns:**

["NO or VERY MINOR ethics concerns only"]

**Final Justification:**

The authors addressed most of my concerns and, I think, most of the other reviewers'. They added new experiments and were thorough. I keep my score and recommend the acceptance of the paper.

**Limitations:**

The authors adequately addressed the limitations.

**Quality:**

4

**Strengths And Weaknesses:**

__Strengths:__
- The authors propose an original score that is needed and will be useful to researchers in causal representation learning.
- The paper is clear and well-written.

__Weakness:__
- The empirical validation of the score compared to existing methods is limited and could be more convincing.

---

> ### Author Rebuttal · Authors · 2025-07-30
>
> We sincerely appreciate your positive feedback and would like to address your questions point by point.
>
> &nbsp;
>
> ## Additional Experimental Results
>
> **Table 4: Spearman correlation coefficients for Section 5.1**
>
> |  Model 	| $\\mathbf{Z}\_1$      	            | $\\mathbf{Z}\_2$      	            | $\\mathbf{Z}\_3$      		|
> |:---:|---------------------|---------------------|---------------------|
> | A 	| $1.0000 \\pm 0.0000$      	            | $0.8568 \\pm 0.0044$ 	| $0.8864 \\pm 0.0040$ 	|
> | B 	| $0.8434 \\pm 0.0061$ 	| $0.9602 \\pm 0.0017$ 	| $0.9908 \\pm 0.0004$ 	|
> | C 	| $0.9673\\pm 0.0013$  	| $0.7215 \\pm 0.0076$ 	| $0.8016 \\pm 0.0062$ 	|
>
> * **Experimental Setup**: The results are computed using the same datasets as in **`Section 5.1`**,  where the representation $\\widehat{\\mathbf{Z}}\_{A\_1}$ is supposed to exclusively measure $\\mathbf{Z}\_1$. Note that the learned representation has dimensionality 1, while the latent space has dimensionality 3\. Therefore, we compute correlation coefficients individually, i.e., \\( \\text{corrcoef}(\\widehat{\\mathbf{Z}}\_{A\_1}, \\mathbf{Z}\_i) \\) for \\( i \= 1, 2, 3 \\).
> * **Results:** As shown, similar to \\( R^2 \\), the correlation coefficients fail to provide a clear correspondence between the representation and the latent variables. This illustrates why the Mean Correlation Coefficient (MCC) can yield deceptively high scores, even when the learned representation is misaligned with the identifiability theory.
>
> &nbsp;
>
> ## Response to Weaknesses
>
> ***W1: Empirical validation of the score compared with other existing methods***
>
> * Thank you for this suggestion. We have included several additional results to further validate our approach, including **`Table 4`** (evaluating MCC), **`Table 3`** (comparison to self-compatibility score, see rebuttal for Reviewer **`k71J`**), and **`Table 5`** (comparison to $R^2$ and MCC under weak and nonlinear causal relations, see rebuttal for Reviewer **`yCFv`**).
> * We deeply apologize that we can not include these results here due to the character limit. Please refer to other rebuttals for more explanation.
>
> &nbsp;
>
> ## Response to Questions
>
> ***Q1: Usefulness of the measurement model under one-to-one correspondence of representations***
>
> * Thank you for this thoughtful question. We agree that many CRL methods offer theoretical guarantees of identifiability up to permutation and scaling. However, in practice, such guarantees often **do not fully translate** into the learned representations due to **finite sample sizes**, **model misspecification**, or **optimization issues**.
> * For example, theory may guarantee that
> \\[\\widehat{\\mathbf{Z}}\_1 \= a \\cdot \\mathbf{Z}\_1,\\quad \\widehat{\\mathbf{Z}}\_2 \= b \\cdot \\mathbf{Z}\_2 \\quad \\text{with } a, b \\neq 0,\\] yet in practice, one might observe entangled representations such as \\[\\widehat{\\mathbf{Z}}\_1 \= \\mathbf{Z}\_1 \+ \\mathbf{Z}\_2,\\quad \\widehat{\\mathbf{Z}}\_2 \= \\mathbf{Z}\_2 \\]
>
>    This leads to a discrepancy between: **(a)** the *ideal measurement model* induced by the identifiability theory (a one-to-one correspondence between latents and measurements), and **(b)** the *empirical measurement model* reflected by the learned representation (where $\\widehat{\\mathbf{Z}}\_1$ mixes both $\\mathbf{Z}\_1$​ and $\\mathbf{Z}\_2$​​). **T-MEX is designed to quantify exactly this kind of misalignment.**
>
> ***Q2: Compare T-MEX and MCC in the experiments.***
> Thanks for your suggestion. We include this additional result in **`Table 4`** and will update our manuscript accordingly.
>
> ***Q3: Discrete nature of T-MEX***
> Thank you for this insightful observation. You're absolutely right that T-MEX, by design, focuses on **exclusivity violations in a discrete manner**,  indicating whether a learned representation incorrectly encodes information from unintended causal variables. However, we would like to offer a few clarifications:
>
> * While T-MEX itself returns a discrete score (counting conditional independence violations), the underlying **CI tests produce p-values**, which provide a *graded measure of evidence* against the null. This enables practitioners to assess the *strength* of each exclusivity relation beyond a binary decision.
>
> * The **discrete nature of T-MEX is similar to metrics like Structural Hamming Distance (SHD)** used in causal discovery, which also quantify edge-wise structural mismatches without measuring correlation strength. We see this as complementary to *predictivity-focused metrics*.
>
> * In fact, **T-MEX and predictivity-focused metrics like $R^2$ or MCC  can and should be used together**. While T-MEX evaluates whether the learned structure aligns with the expected measurement model (from a *causal* perspective), metrics such as $R^2$ and MCC assess *how well* the representation predicts the ground-truth latent variable (from a *predictive* **non-causal** perspective).
>
> * As noted in **`Remark 3.2`**, the measurement model can be further extended to incorporate additional assumptions, such as combining exclusivity checks with constraints on correlation strength or parametric function classes.
>
> ***Q4:*** ***$R^2$ interpretation in Section 5.1*****:**
>
> * While the average $R^2$ scores happen to match the T-MEX ordering in Section 5.1, this is coincidental and not generally reliable.
> * If a model is expected to disentangle $\\mathbf{Z}\_1$​, one might look at its $R^2$ score on $\\mathbf{Z}\_1$​. However, **Model C** shows a **high R² on $\\mathbf{Z}\_1$**, even though its representation additionally mixes **$ \\mathbf{Z}\_2$​ and $\\mathbf{Z}\_3$​**, leading to substantial ATE bias (**`Figure 3`** in **`Section 5.1`**). This misalignment is correctly flagged by T-MEX but not by $R^2$.
> * More broadly, $R^2$ **does not indicate which latent variable is being captured** by each representation dimension, and it is unclear how to rank models consistently when representations are entangled. **T-MEX addresses this by explicitly testing measurement exclusivity**, making it a more principled tool for causal representation evaluation.
>
> **Q5: Interpretation of the measurement model for the ISTAnt experiment (Section 5.2):**
>
> * The **videos are generated by both the treatment $\\mathbf{T}$**, the **outcome $\\mathbf{Y}$, and other experimental settings that we do not see, but may correlate the predictions with the treatment (not $\\mathbf{Y}$, as we have a Randomized Controlled Trial).** Since we focus only on models with **classification accuracy above 0.8** (see **`Figure 5`** in **`Section 5.2`**), the learned representation $\\widehat{\\mathbf{Y}}$ must encode meaningful information about $\\mathbf{Y}$,  ruling out the trivial “empty graph” case. However, we need to test whether the predictions contain spurious correlations with the treatment assignment $\mathbf{T}$, which T-MEX captures (see **`Appendix D.2`** for more details).
>
> * This leads to two relevant causal structures:
>
>   1\.  \\( T \\rightarrow \\widehat{Y} \\leftarrow Y \\): the representation is entangled with both treatment and outcome (T-MEX \= 1),
>   2\. \\( Y \\rightarrow \\widehat{Y} \\): the representation exclusively captures the outcome (T-MEX \= 0).
>
> * These correspond exactly to the two T-MEX levels we analyze. As shown in **`Figure 5`**, T-MEX discriminates between models with high and low ATE bias and classification performance, offering a meaningful evaluation for real-world scientific applications.
>
> Thanks for pointing out the typos. We will correct them in the updated manuscript.

---

### Official Review · Reviewer_k71J · 2025-06-28

**Clarity:** 3
**Significance:** 2
**Originality:** 3
**Rating:** 4
**Confidence:** 4

**Summary:**

The paper treats causal-representation learning as a measurement problem: each learned code dimension should be an exclusive, CI-verifiable proxy for a single latent cause. It proposes T-MEX, a conditional-independence score that quantifies this exclusivity and proves a finite-sample link between low T-MEX and accurate causal-effect estimates. Simulations and an ecological-video benchmark show T-MEX tracks ATE bias better than common metrics like $R^2$ and MCC.

**Questions:**

1. Can you compare T-MEX against CI-based graph validation metrics (for example, self-compatibility (von Kügelgen ’21; Ahuja ’23) or falsification/compatibility score (Marx & Vreeken ’22)) on the same representations?
2. Would other CI tests give consistent rankings? Can you verify with experiments?
3. Can you analyze the trade-off between T-MEX and the size of the CI oracle? Something like a worst-case or average-case bound on the number of CI tests T-MEX performs as a function of dimensions would suffice?
4. Can you provide a recommendation of the $\alpha$-level and sample-size heuristics?
5. Is it possible to scale the experiments to aournd 20 latent variables with 100 observable variables? How does the runtime evolve witht he dimensions?

I would be happy to raise my socre if my questions can be addressed.

**Ethical Concerns:**

["NO or VERY MINOR ethics concerns only"]

**Final Justification:**

The authors have addressed most of my concerns during the rebuttal. Some minor points are partially addressed, which is reasonable. I believe the additional discussions and experiments will greatly improve the paper.

**Limitations:**

yes

**Paper Formatting Concerns:**

There is no formatting concerns.

**Quality:**

3

**Strengths And Weaknesses:**

Strengths:
1. This paper formalises CRL as a measurement model and provides sound theoretical analysis.
2. The authors conduct wll-designed small-scale studies.
3. T-MEX is algorithm-agnostic and needs only observational samples, potentially valuable where ground-truth ATE is unavailable. It connects identification theory with downstream estimands via the notion of causal validity of a measurement model.

Weaknesses:
My major concern is that the main contribution of T-MEX is that it can be potentially used to evaluate CRL methods when we know the ground truth causal structure and variables. While in the current paper, there is no explicit guidance on how to conduct this. For a random CRL method, how would T-MEX automatically find the block matching? Can T-MEX surrogate be used if we do not have access to the true variables? How would the grid of the CI test affect the performance? Is T-MEX efficient for large block sizes?
Here are some minor concerns.
1. There are multiple typos in the current version (for example, 'diffeomporphism' in line 121).
2. The empirical evaluations are on small datasets. The simulations have only 5 variables.
3. T-MEX relies on a valid, powerful CI oracle which may hinder its utility in real-world applications.
4. The evaluations do not show the confidence intervals for T-MEX.
5. The reliacne on the test choice is not shown.
6. The relation between T-MEX and recent works on self-compatibility/falsification can be discussed more in detail in section 4 to highlight the difference.

---

> ### Author Rebuttal · Authors · 2025-07-30
>
> Thank you for your thoughtful comments and constructive feedback. We address your questions below.
>
> &nbsp;
>
> ## Additional Experimental Results
>
> **T-MEX Oracle:** T-MEX score computed based on an oracle CI test with 0 type I error and power 1.
>
> **Table 1**: Due to character limit, we kindly invite you to find this table in the rebuttal for Reviewer **`yCFv`**.
>
> **Table 2: T-MEX under different CI Tests**
>
> | CI test  |  Model |       T-MEX          |
> |:---------:|:-------:|---------------------:|
> | PCM (RF) |    A   | $0.0000 \\pm 0.0000$    |
> | PCM (RF) |    B   | $0.8000 \\pm 0.7559$    |
> | PCM (RF) |    C   | $2.0000 \\pm 0.0000$    |
> | KCI      |    A   | $0.0400 \\pm 0.1979$    |
> | KCI      |    B   | $0.1600 \\pm 0.4218$    |
> | KCI      |    C   | $2.0000 \\pm 0.0000$    |
>
> * **Experimental Setup:** We follow the same settings in **`Section 5.1`**. We consider PCM test (see **`Appendix E)`** with random forest (RF) and Kernel Conditional Independence (KCI) test **`[Zhang et al. 2012]`**.
> * **Results:** T-MEX ranks the models consistently with the results in **`Section 5.1`**.
>
> *Zhang, K., Peters, J., Janzing, D., & Schölkopf, B. (2011). Kernel-based conditional independence test and application in causal discovery. In Proceedings of the Twenty-Seventh Conference on Uncertainty in Artificial Intelligence (pp. 804-813).*
>
> **Table 3: Comparison of T-MEX and self-incompatibility (SI) score \[Faller et al., 2024\] using three causal discovery methods (PC, FCI, RCDLINGAM)**
>
> |T-MEX Oracle | T-MEX 	| SI score (PC) 	| SI score (FCI) 	| SI score (RCDLINGAM) 	|
> |:----------------:|-------:|:-------------:|:--------------:|:--------------------:|
> |         1             | $1.1000 \\pm 0.3606$   	| $0.0 \\pm 0.0$ 	| $0.0 \\pm 0.0$  | $0.0 \\pm 0.0$      |
>
> * **Experimental Setup**: We follow the example in **`Appendix D.3`** with 3 causal variables and 3 measurements. Specifically, \\( \\widehat{\\mathbf{Z}}\_2 \\) and \\( \\widehat{\\mathbf{Z}}\_3 \\) correspond to \\( \\mathbf{Z}\_2 \\) and \\( \\mathbf{Z}\_3 \\) respectively, while \\( \\widehat{\\mathbf{Z}}\_1 \\) remains a ***mixture*** of \\( \\mathbf{Z}\_1 \\) and \\( \\mathbf{Z}\_2 \\)*,* as shown in **`Figure 8`**. We assume the **element-wise identification as the ground-truth measurement model** (a common identification guarantee by many CRL methods).
> * **Results**: T-MEX correctly detects exclusivity violations, consistently yielding a score close to the oracle 1 . **In contrast, the unsupervised SI score fails to identify the mixing effect, assigning zero incompatibility even when the representation is entangled.**
>
> *Faller, P. M., Vankadara, L. C., Mastakouri, A. A., Locatello, F., & Janzing, D. (2024). Self-compatibility: Evaluating causal discovery without ground truth. In International Conference on Artificial Intelligence and Statistics (pp. 4132-4140). PMLR.*
>
> &nbsp;
>
> ## Response to Weaknesses
>
> ***W1: Assumption on knowing the ground truth measurement model***
>
> * **T-MEX does not require knowledge of the causal graph among latent variables.** Instead, it only assumes a known correspondence between each ground-truth causal variable and its intended measurement .
> * **Assuming knowledge about the latent variables aligns with common practice in CRL evaluation**. Metrics like $R^2$ and MCC also rely on access to the ground-truth latent variables for benchmarking.
> * **What sets T-MEX apart is its ability to account for causal dependencies** between latent variables — a key difference from disentanglement metrics, which typically assume independent latent factors. This makes the evaluation more challenging but also more aligned with real-world causal tasks.
> * **T-MEX is not designed for validation/model selection, but it can be used this way if supervision is available.** This is reasonable especially in structured scientific settings (e.g., randomized trials, task-specific modeling), where it is clear which variables must be measured for downstream causal reasoning. We demonstrated this in **`Section 5.2`** with a real-world ecology causal task.
>
> ***W2: Block-matching***
>
> * T-MEX can provide the matching between each learned representation block and the underlying causal variables. Importantly, **it does *not* require a fixed index alignment and can learn the matching up to permutations when there is an element-wise correspondence between the measurements and the causal variables,**  as we explain next.
> * For a random CRL method where the indexing of representation blocks is arbitrary, **one can compute the T-MEX score after constructing the empirical measurement model under all valid permutations of the latents and select the assignment that yields the *lowest* score.** This best-matching procedure ensures that T-MEX reflects the true degree of exclusivity, regardless of how the CRL method arranges its outputs.
> * This is analogous to how existing metrics (e.g., MCC) are typically computed under the best matching permutation between latent and learned components.
>
>
> ***W3: T-MEX with large latent dimensions***
>
>
> * We fully agree that CI testing in large dimensions is a well-known statistical challenge. In our experiments in **`Section 5`**, we employ the PCM test which is theoretically solid and implemented efficiently in the **`pycomets`** package. We show in **`Table 1`** that T-MEX remains computationally tractable and efficient up to **50 causal variables and 50 measurement variables**.
> * That said, we would like to emphasize that **an extremely large number of latent causal variables is often undesirable in practice**. Most real-world applications naturally decompose into a small number of interpretable causal factors. Accordingly, the majority of existing CRL benchmarks and methods focus on **small- to medium-scale latent spaces** (e.g., 5–20), which is also the regime where downstream causal reasoning (e.g., ATE estimation, adjustment) is most tractable. We agree, however, that this is a limitation, and we encourage future works in this direction (e.g., moving to ML-driven dependence measures that may be less reliant on dimensions).
>
> Response to minor concerns
>
> 1. **Typos***:* Thank you for carefully reviewing our paper. We will correct them in the new version.
> 2. **Evaluate on a larger dataset***:* **`Table 1`** shows additional T-MEX experiments with an increasing number of latents.
> 3. **Reliance on CI tests**:
>     * We fully agree that **CI testing is an important component** in our framework, and choosing a valid and powerful test is essential.  We chose PCM test in our experiments for its theoretical guarantees on the validity (type I error control) and power of the test, making it a reliable choice of test for the exclusivity claim.
>     * Despite known challenges, CI testing remains a **flexible and principled approach for evaluating exclusivity**—an aspect that standard metrics like MCC or $R^2$ cannot capture under causal dependencies. We will add this discussion to the paper.
>
>
> 4. **Confidence intervals for T-MEX:**
>    * **`Table 1` `in the main paper`** shows the standard deviation of T-MEX over 50 different simulations.
>    * Additionally, **`Figure 6`** in **`Appendix D.1`** reports the p-values for **`Section 5.1`**, which indicates the confidence level of all individual tests that result in the T-MEX score.
>
> 5. **Reliance on the test choice**: **`Table 2`** shows that T-MEX remains consistent across different test choices.
> 6. **Related work on self-compatibility/falsification:**
>    Self-incompatibility (SI) score **`[Faller at al. 2024]`** is an unsupervised score for measuring inconsistency between different subgraphs quantified by SHD (We chose to compare with this work as it is, to the best of our knowledge, the most recent one regarding self-compatibility), and thus  can be viewed as an unsupervised proxy of SHD. However, as we show in **`Table 3`** and **`Appendix D.3`**, both SI and SHD failed to detect mixing in the learned representations.
>
> &nbsp;
>
> ## Response to Questions
>
> ***Q1: Compare T-MEX with other graph validation metrics***
> **`Table 3`** evaluates **T-MEX and the self-incompatibility (SI) score proposed by Faller et al. (2024)**.
>
> ***Q2: T-MEX scores under different CI tests***
> **`Table 2`** presents additional results for the simulation in **`Section 5.1`** using different CI tests.
>
> ***Q3: Number of CI tests as a function of dimensions***
>
> * For $N$ causal variables and $M$ measurement blocks, we need $M \\times N$ tests; also see **`Algorithm 1`** in the **`Appendix`**.
> * **`Table 1`** provides a **runtime analysis** for different numbers of latents to show the practical applicability of T-MEX.
>
> ***Q4: Recommendation of the alpha-level and sample-size heuristics***
> T-MEX compares the adjacency matrix implied by the measurement model to the one estimated from data using CI tests, both false positives and false negatives can affect the score.
>
> * For valid CI tests such as PCM and GCM (see **`Appendix E`**), larger sample sizes generally improve the power of the tests.
> * Moreover, there is a trade-off between power and false positive rate: at fixed sample size, increasing alpha improves power but also raises the Type I error rate.
> * **We recommend choosing alpha and sample size based on the sparsity of the measurement model**: in sparser settings, controlling false positives is more important, and a smaller alpha is preferable. For practical guidance, we refer to recommendations in statistical testing, such as **`[Lakens 2022]`**.
>
> *Lakens, D., 2022. Sample size justification. *Collabra: psychology*, *8*(1), p.33267.*
>
> ***Q5: Scaling up experiments***
>
> * We would like to clarify, **T-MEX evaluates the correspondence between learned representations and ground-truth latent variables**; it does not perform representation learning or dimensionality reduction itself.
> * Following your suggestion, we report additional results in **`Table 1`**.

---

> > ### Comment · Reviewer_k71J · 2025-08-05
> >
> > I thank the authors for addressing most of my concerns. Accordingly, I will raise my rating to 4.

---

> > > ### Author Response · Authors · 2025-08-07
> > >
> > > We appreciate you taking the time to review our responses. We are pleased to hear that our responses addressed your concerns and that your assessment of the paper has improved. We’ll make sure to incorporate the points discussed into our revision. Thank you again for your thoughtful comments.

---

### Official Review · Reviewer_ns4w · 2025-07-03

**Clarity:** 3
**Significance:** 2
**Originality:** 3
**Rating:** 4
**Confidence:** 3

**Summary:**

The authors recast causal representation learning (CRL) as a third pillar of causal analysis by treating learned latent codes as measurements of hidden causal variables. Within this formal “measurement-model” framework they introduce T-MEX—Test-based Measurement EXclusivity—an evaluation score that counts conditional-independence violations of the exclusivity relations implied by a chosen measurement model. They prove finite-sample bounds showing T-MEX is controlled by the Type-I / Type-II error of the underlying CI tests and empirically demonstrate that the score predicts downstream ATE bias far better than correlation- or disentanglement-based metrics.

**Questions:**

See my comments above.

**Ethical Concerns:**

["NO or VERY MINOR ethics concerns only"]

**Final Justification:**

I have read the response. I am slightly leaning towards accepting the paper.

**Limitations:**

See my comments above.

**Paper Formatting Concerns:**

No.

**Quality:**

3

**Strengths And Weaknesses:**

**Strengths**

- The paper is well-written and easy to follow.
- The paper thoroughly explains why previous evaluation metrics and disentanglement scores can be misleading when latent variables are causally related.
- The experimental results are interesting and solid.


**Weaknesses**

- **Strong reliance on exclusivity alone.** IIUC, the framework checks only whether any pair of learned features violates exclusivity given the others; it does not test other measurement-model axioms such as invertibility, monotonicity, or functional form. A representation that passes T-MEX could still be useless for causal inference if, for example, several causal variables collapse into a single latent factor.

- **Linear-Gaussian focus** and **sample-complexity limits of CI testing.** All theoretical derivations assume additive Gaussian error terms (for analytic tractability of CI tests) and all simulations instantiate linear DAGs. Real-world problems often involve nonlinear, heteroskedastic, or discrete mechanisms; whether T-MEX still correlates with causal-effect bias under such violations is unknown. Each exclusivity claim is evaluated via a multivariate conditional-independence (CI) test whose power decays quickly with dimensionality and conditioning-set size. While Proposition 3.1 upper-bounds the expected T-MEX under the null, the paper does not translate this into concrete guidance on how many samples are needed for typical vision or language embeddings with hundreds of features, leaving practitioners without a way to judge feasibility.

I am less familiar with the current progress in causal representation learning, so any clarifications on the questions above would help me to understand the background better.

---

> ### Author Rebuttal · Authors · 2025-07-30
>
> We thank the reviewer for their time and valuable feedback. We will address the remaining concerns as follows:
>
> &nbsp;
>
> ## Additional Experimental Results
>
> **Table 7: T-MEX v.s. ATE bias under nonlinear SCM**
>
> |  Model  |        T-MEX      |  abs(ATE bias)    |
> | -------:|------------------:|------------------:|
> | Perfect |$0.0100 \\pm 0.1000$| $0.1532 \\pm 0.0283$|
> |  Mixed  |$1.0000 \\pm 0.0000$| $0.5357 \\pm 0.0540$|
>
> * **Experimental Setup:** We consider the same causal graph as given in **`Section 5.1`**, where $\\mathbf{Z}\_1$ confounds $\\mathbf{Z}\_2$ and $\\mathbf{Z}\_3$, and needed to be adjusted for a valid treatment effect estimation. A *perfect* model means $\\widehat{\\mathbf{Z}}\_1$ exclusively measures $\\mathbf{Z}\_1$, whereas a *mixed* model indicates an entangled representation, i.e.,  *$\\widehat{\\mathbf{Z}}\_1$ mixes $\\mathbf{Z}\_1$ and $\\mathbf{Z}\_2$*
> * **Results:** Similar to the linear case (**`Section 5.1`**), T-MEX closely aligns with the absolute values of the ATE bias, effectively evaluating causal representations for downstream inference tasks with nonlinear causal relations.
>
> &nbsp;
>
> ## Response to Weaknesses
>
> ***W1: Strong reliance on exclusivity alone.***
>
> * Thank you for this thoughtful comment\! We fully agree that additional properties of the measurement functions $h\_{ij}$, such as invertibility or parametric functional form, are important for further investigating the causal validity of a representation for a specific causal task.
> * As discussed in **`Remark 3.2`, our framework intentionally operates in a non-parametric setting to allow broad applicability, agnostic to the form of** $h\_{ij}$​. That said, the framework is easily extensible: once exclusivity is verified, one may perform additional tests for properties like linear or nonlinear invertibility (e.g., via $R^2$ or correlation metrics) on the detected latent-measurement pairs. **Since the form of $h\_{ij}$ can vary across use cases (e.g., diffeomorphism or affine transformations), we chose not to include such tests in the general-purpose T-MEX score**.
> * Importantly, **exclusivity serves as a necessary foundation for evaluating such higher-order properties.** Without a reliable one-to-one correspondence between latent variables and learned representations, questions of invertibility or functional form become ill-posed.
> * However, we fully agree that for a specific causal downstream task, it is important to evaluate both exclusivity and additional functional constraints such as invertibility. This is why **we also report high $R^2$ (`Section 5.1`) and classification accuracy (`Section 5.2`) to confirm that the learned representations retain enough task-relevant information beyond structural exclusivity.** We will add a paragraph to the paper to clarify this point further.
>
> *“A representation that passes T-MEX could still be useless for causal inference if, for example, several causal variables collapse into a single latent factor.”*
>
> There are a few subtleties about this statement in addition to the discussion on the measurement function above.
>
> The measurement model details which causal inference questions can be answered from the measurements, and T-MEX tests alignment between the representation and the model. We interpret the reviewer's concern as that T-MEX may pass, but the resulting measurement model is still insufficient for certain causal inference estimands. In our view, this is a **modeling issue, and not an evaluation issue.** In this case, one needs to refine the measurement model (e.g., by collecting more data with specific interventions) and train a new causal representation learning method that allows identification of a more fine-grained model.
>
> ***W2: Linear-Gaussian focus and sample-complexity limits of CI testing.***
>
> ***T-MEX with nonlinear SCM***
>
> * Thank you very much for suggesting this new experiment. We include new experimental results with nonlinear SCM, following the causal graph structure in **`Section 5.1`** (see **`Table 7`**).
> * As shown by the results, **the T-MEX score also closely aligns with the absolute ATE bias under non-linear SCM**, showcasing the general applicability of T-MEX under a wide range of CRL scenarios.
>
>
> ***Sample-complexity limits of CI testing***
>
> * The sample complexity of T-MEX depends on the sample complexity of the chosen CI test. For the Projected Covariance Measure (PCM) test we used in our experiments, we refer to **`Theorems 5 and 7`** in **`[Lundborg et al., 2024]`** for asymptotic analysis. The theorems show that the power of the test converges to 1 uniformly for a set of alternatives that are sufficiently separated from the null distributions. We will more explicitly acknowledge the challenges of conditional independence tests, but we remark that this is an active research field in statistics and our score is agnostic to the specific test that is used.
> * Regarding your concern about the feasibility in vision or language settings, it is crucial to distinguish between the latent causal variables—whose identifiability is the goal of CRL—and the typically high-dimensional learned embeddings from raw data. Most real-world applications can be framed around a small number of causal variables that are descriptive for a specific downstream task. These can be predicted or extracted from high-dimesnional representations learned by foundation models (see **`Section 5.2`** for a real-world example).
>
> *A. R. Lundborg, I. Kim, R. D. Shah, and R. J. Samworth. The projected covariance measure for assumption-lean variable significance testing. The Annals of Statistics, 52(6):2851–2878, 2024.*

---

> > ### Comment · Reviewer_ns4w · 2025-08-05
> > **Thanks for your response.**
> >
> > I would suggest the authors to include new results in the next revision. I am maintaining my previous scores.

---

> > > ### Author Response · Authors · 2025-08-07
> > >
> > > Thank you for taking the time to review our responses. We appreciate your thoughtful feedback and constructive suggestions, and we are glad to hear that you continue to view the paper favorably. We will be sure to incorporate the points discussed during the rebuttal into the revision.

---

### Decision · Program_Chairs · 2025-09-17

**Decision:**

Accept (poster)

**Comment:**

The paper studies the role of causal representation learning and introduces a new evaluation metric, the Test-based Measurement EXclusivity (T-MEX) score. This score is designed to assess how well learned representations reflect the true causal structure. The proposal is validated through both theory and experiments with synthetic data and real-world video analysis.

The reviewers appreciated the exposition of the paper and choice of experiments. The T-MEX score is a clear and interpretable quantity that is of relevance to a large number of researchers in the field.

The discussion phase led to the authors conducting a number of new experiments, with noisy CI oracles, non-linear SCM's, increased number of latents, and with noisy data. The reviewers appreciated the new information, and we'd encourage including these findings in the final revision.